# The Multiple Quantile Graphical Model

**Alnur Ali**
Machine Learning Department
Carnegie Mellon University
alnurali@cmu.edu

**J. Zico Kolter**
Computer Science Department
Carnegie Mellon University
zkolter@cs.cmu.edu

**Ryan J. Tibshirani**
Department of Statistics
Carnegie Mellon University
ryantibs@cmu.edu

## Abstract

We introduce the Multiple Quantile Graphical Model (MQGM), which extends the *neighborhood selection* approach of Meinshausen and Bühlmann for learning sparse graphical models. The latter is defined by the basic subproblem of modeling the conditional mean of one variable as a sparse function of all others. Our approach models a set of conditional quantiles of one variable as a sparse function of all others, and hence offers a much richer, more expressive class of conditional distribution estimates. We establish that, under suitable regularity conditions, the MQGM identifies the exact conditional independencies with probability tending to one as the problem size grows, even outside of the usual homoskedastic Gaussian data model. We develop an efficient algorithm for fitting the MQGM using the alternating direction method of multipliers. We also describe a strategy for sampling from the joint distribution that underlies the MQGM estimate. Lastly, we present detailed experiments that demonstrate the flexibility and effectiveness of the MQGM in modeling hetereoskedastic non-Gaussian data.

## 1   Introduction

We consider modeling the joint distribution $\mathbf{Pr}(y_1, \ldots, y_d)$ of $d$ random variables, given $n$ independent draws from this distribution $y^{(1)}, \ldots, y^{(n)} \in \mathbf{R}^d$, where possibly $d \gg n$. Later, we generalize this setup and consider modeling the conditional distribution $\mathbf{Pr}(y_1, \ldots, y_d | x_1, \ldots, x_p)$, given $n$ independent pairs $(x^{(1)}, y^{(1)}), \ldots, (x^{(n)}, y^{(n)}) \in \mathbf{R}^{p+d}$. Our starting point is the *neighborhood selection* method [28], which is typically considered in the context of multivariate Gaussian data, and seen as a tool for *covariance selection* [8]: when $\mathbf{Pr}(y_1, \ldots, y_d)$ is a multivariate Gaussian distribution, it is a well-known fact that $y_j$ and $y_k$ are conditionally independent given the remaining variables if and only if the coefficent corresponding to $y_k$ is zero in the (linear) regression of $y_j$ on all other variables (*e.g.*, [22]). Therefore, in neighborhood selection we compute, for each $k = 1, \ldots, d$, a lasso regression — in order to obtain a small set of conditional dependencies — of $y_k$ on the remaining variables, *i.e.*,

$$\underset{\theta_k \in \mathbf{R}^d}{\text{minimize}} \ \sum_{i=1}^{n} \left( y_k^{(i)} - \sum_{j \neq k} \theta_{kj} y_j^{(i)} \right)^2 + \lambda \|\theta_k\|_1, \tag{1}$$

for a tuning parameter $\lambda > 0$. This strategy can be seen as a *pseudolikelihood* approximation [4],

$$\mathbf{Pr}(y_1, \ldots, y_d) \approx \prod_{k=1}^{d} \mathbf{Pr}(y_k | y_{\neg k}), \tag{2}$$

where $y_{\neg k}$ denotes all variables except $y_k$. Under the multivariate Gaussian model for $\mathbf{Pr}(y_1, \ldots, y_d)$, the conditional distributions $\mathbf{Pr}(y_k | y_{\neg k})$, $k = 1, \ldots, d$ here are (univariate) Gaussians, and maximizing the pseudolikelihood in (2) is equivalent to separately maximizing the conditionals, as is precisely done in (1) (with induced sparsity), for $k = 1, \ldots, d$.

Following the pseudolikelihood-based approach traditionally means carrying out three steps: (i) we write down a suitable family of joint distributions for $\mathbf{Pr}(y_1, \ldots, y_d)$, (ii) we derive the conditionals $\mathbf{Pr}(y_k|y_{\neg k})$, $k = 1, \ldots, d$, and then (iii) we maximize each conditional likelihood by (freely) fitting the parameters. Neighborhood selection, and a number of related approaches that came after it (see Section 2.1), can be all thought of in this workflow. In many ways, step (ii) acts as the bottleneck here, and to derive the conditionals, we are usually limited to a homoskedastic and parameteric family for the joint distribution.

The approach we take in this paper differs somewhat substantially, as we *begin* by directly modeling the conditionals in (2), without any preconceived model for the joint distribution — in this sense, it may be seen a type of *dependency network* [13] for continuous data. We also employ heteroskedastic, nonparametric models for the conditional distributions, which allows us great flexibility in learning these conditional relationships. Our method, called the Multiple Quantile Graphical Model (MQGM), is a marriage of ideas in high-dimensional, nonparametric, multiple quantile regression with those in the dependency network literature (the latter is typically focused on discrete, not continuous, data).

An outline for this paper is as follows. Section 2 reviews background material, and Section 3 develops the MQGM estimator. Section 4 studies basic properties of the MQGM, and establishes a structure recovery result under appropriate regularity conditions, even for heteroskedastic, non-Gaussian data. Section 5 describes an efficient ADMM algorithm for estimation, and Section 6 presents empirical examples comparing the MQGM versus common alternatives. Section 7 concludes with a discussion.

## 2 Background

### 2.1 Neighborhood selection and related methods

Neighborhood selection has motivated a number of methods for learning sparse graphical models. The literature here is vast; we do not claim to give a complete treatment, but just mention some relevant approaches. Many pseudolikelihood approaches have been proposed, see *e.g.*, [35, 33, 12, 24, 17, 1]. These works exploit the connection between estimating a sparse inverse covariance matrix and regression, and they vary in terms of the optimization algorithms they use and the theoretical guarantees they offer. In a clearly related but distinct line of research, [45, 2, 11, 36] proposed $\ell_1$-penalized likelihood estimation in the Gaussian graphical model, a method now generally termed the *graphical lasso* (GLasso). Following this, several recent papers have extended the GLasso in various ways. [10] examined a modification based on the multivariate Student $t$-distribution, for robust graphical modeling. [37, 46, 42] considered conditional distributions of the form $\mathbf{Pr}(y_1, \ldots, y_d|x_1, \ldots, x_p)$. [23] proposed a model for mixed (both continuous and discrete) data types, generalizing both GLasso and pairwise Markov random fields. [25, 26] used copulas for learning non-Gaussian graphical models.

A strength of neighborhood-based (*i.e.*, pseudolikelihood-based) approaches lies in their simplicity; because they essentially reduce to a collection of univariate probability models, they are in a sense much easier to study outside of the typical homoskedastic, Gaussian data setting. [14, 43, 44] elegantly studied the implications of using univariate exponential family models for the conditionals in (2). Closely related to pseudoliklihood approaches are dependency networks [13]. Both frameworks focus on the conditional distributions of one variable given all the rest; the difference lies in whether or not the model for conditionals stems from first specifying some family of joint distributions (pseudolikelihood methods), or not (dependency networks). Dependency networks have been thoroughly studied for discrete data, *e.g.*, [13, 29]. For continuous data, [40] proposed modeling the mean in a Gaussian neighborhood regression as a nonparametric, additive function of the remaining variables, yielding flexible relationships — this is a type of dependency network for continuous data (though it is not described by the authors in this way). Our method, the MQGM, also deals with continuous data, and is the first to our knowledge that allows for fully nonparametric conditional distributions, as well as nonparametric contributions of the neighborhood variables, in each local model.

### 2.2 Quantile regression

In linear regression, we estimate the conditional mean of $y|x_1, \ldots, x_p$ from samples. Similarly, in $\alpha$-*quantile regression* [20], we estimate the conditional $\alpha$-quantile of $y|x_1, \ldots, x_p$ for a given $\alpha \in [0, 1]$, formally $Q_{y|x_1,\ldots,x_p}(\alpha) = \inf\{t : \mathbf{Pr}(y \leq t|x_1, \ldots, x_p) \geq \alpha\}$, by solving the convex optimization problem: $\text{minimize}_\theta \sum_{i=1}^{n} \psi_\alpha(y^{(i)} - \sum_{j=1}^{p} \theta_j x_j^{(i)})$, where $\psi_\alpha(z) = \max\{\alpha z, (\alpha - 1)z\}$ is the $\alpha$-

quantile loss (also called the "pinball" or "tilted absolute" loss). Quantile regression can be useful when the conditional distribution in question is suspected to be heteroskedastic and/or non-Gaussian, *e.g.*, heavy-tailed, or if we wish to understand properties of the distribution other than the mean, *e.g.*, tail behavior. In multiple quantile regression, we solve several quantile regression problems simultaneously, each corresponding to a different quantile level; these problems can be coupled somehow to increase efficiency in estimation (see details in the next section). Again, the literature on quantile regression is quite vast (especially that from econometrics), and we only give a short review here. A standard text is [18]. Nonparametric modeling of quantiles is a natural extension from the (linear) quantile regression approach outlined above; in the univariate case (one conditioning variable), [21] suggested a method using smoothing splines, and [38] described an approach using kernels. More recently, [19] studied the multivariate nonparametric case (more than one conditioning variable), using additive models. In the high-dimensional setting, where $p$ is large, [3, 16, 9] studied $\ell_1$-penalized quantile regression and derived estimation and recovery theory for non-(sub-)Gaussian data. We extend results in [9] to prove structure recovery guarantees for the MQGM (in Section 4.3).

## 3 The multiple quantile graphical model

Many choices can be made with regards to the final form of the MQGM, and to help in understanding these options, we break down our presentation in parts. First fix some ordered set $\mathcal{A} = \{\alpha_1, \ldots, \alpha_r\}$ of quantile levels, *e.g.*, $\mathcal{A} = \{0.05, 0.10, \ldots, 0.95\}$. For each variable $y_k$, and each level $\alpha_\ell$, we model the conditional $\alpha_\ell$-quantile given the other variables, using an additive expansion of the form:

$$Q_{y_k|y_{\neg k}}(\alpha_\ell) = b_{\ell k}^* + \sum_{j \neq k}^d f_{\ell k j}^*(y_j), \tag{3}$$

where $b_{\ell k}^* \in \mathbf{R}$ is an intercept term, and $f_{\ell k j}^*$, $j = 1, \ldots, d$ are smooth, but not parametric in form. In its most general form, the MQGM estimator is defined as a collection of optimization problems, over $k = 1, \ldots, d$ and $\ell = 1, \ldots, r$:

$$\underset{\substack{b_{\ell k}, \, f_{\ell k j} \in \mathcal{F}_{\ell k j}, \\ j = 1, \ldots, d}}{\text{minimize}} \quad \sum_{i=1}^n \psi_{\alpha_\ell}\left(y_k^{(i)} - b_{\ell k} - \sum_{j \neq k} f_{\ell k j}(y_j^{(i)})\right) + \sum_{j \neq k}\left(\lambda_1 P_1(f_{\ell k j}) + \lambda_2 P_2(f_{\ell k j})\right)^\omega. \tag{4}$$

Here $\lambda_1, \lambda_2 \geq 0$ are tuning parameters, $\mathcal{F}_{\ell k j}$, $j = 1, \ldots, d$ are univariate function spaces, $\omega > 0$ is a fixed exponent, and $P_1, P_2$ are sparsity and smoothness penalty functions, respectively. We give three examples below; many other variants are also possible.

**Example 1: basis expansion model**     Consider taking $\mathcal{F}_{\ell k j} = \mathbf{span}\{\phi_1^j, \ldots, \phi_m^j\}$, the span of $m$ basis functions, *e.g.*, radial basis functions (RBFs) with centers placed at appropriate locations across the domain of variable $j$, for each $j = 1, \ldots, d$. This means that each $f_{\ell k j} \in \mathcal{F}_{\ell k j}$ can be expressed as $f_{\ell k j}(x) = \theta_{\ell k j}^T \phi^j(x)$, for a coefficient vector $\theta_{\ell k j} \in \mathbf{R}^m$, where $\phi^j(x) = (\phi_1^j(x), \ldots, \phi_m^j(x))$. Also consider an exponent $\omega = 1$, and the sparsity and smoothness penalties

$$P_1(f_{\ell k j}) = \|\theta_{\ell k j}\|_2 \quad \text{and} \quad P_2(f_{\ell k j}) = \|\theta_{\ell k j}\|_2^2,$$

respectively, which are group lasso and ridge penalties, respectively. With these choices in place, the MQGM problem in (4) can be rewritten in finite-dimensional form:

$$\underset{b_{\ell k}, \, \theta_{\ell k} = (\theta_{\ell k 1}, \ldots, \theta_{\ell k d})}{\text{minimize}} \quad \psi_{\alpha_\ell}\left(Y_k - b_{\ell k}\mathbf{1} - \Phi\theta_{\ell k}\right) + \sum_{j \neq k}\left(\lambda_1\|\theta_{\ell k j}\|_2 + \lambda_2\|\theta_{\ell k j}\|_2^2\right). \tag{5}$$

Above, we have used the abbreviation $\psi_{\alpha_\ell}(z) = \sum_{i=1}^n \psi_{\alpha_\ell}(z_i)$ for a vector $z = (z_1, \ldots, z_n) \in \mathbf{R}^n$, and also $Y_k = (y_k^{(1)}, \ldots, y_k^{(n)}) \in \mathbf{R}^n$ for the observations along variable $k$, $\mathbf{1} = (1, \ldots, 1) \in \mathbf{R}^n$, and $\Phi \in \mathbf{R}^{n \times dm}$ for the basis matrix, with blocks of columns to be understood as $\Phi_{ij} = \phi(y_j^{(i)})^T \in \mathbf{R}^m$.

The basis expansion model is simple and tends to work well in practice, so we focus on it for most of the paper. In principle, essentially all our results apply to the next two models we describe, as well.

**Example 2: smoothing splines model**     Now consider taking $\mathcal{F}_{\ell k j} = \mathbf{span}\{g_1^j, \ldots, g_n^j\}$, the span of $m = n$ natural cubic splines with knots at $y_j^{(1)}, \ldots, y_j^{(n)}$, for $j = 1, \ldots, d$. As before, we can then write $f_{\ell k j}(x) = \theta_{\ell k j}^T g^j(x)$ with coefficients $\theta_{\ell k j} \in \mathbf{R}^n$, for $f_{\ell k j} \in \mathcal{F}_{\ell k j}$. The work of [27], on high-dimensional additive smoothing splines, suggests a choice of exponent $\omega = 1/2$, and penalties

$$P_1(f_{\ell k j}) = \|G^j \theta_{\ell k j}\|_2^2 \quad \text{and} \quad P_2(f_{\ell k j}) = \theta_{\ell k j}^T \Omega^j \theta_{\ell k j},$$

for sparsity and smoothness, respectively, where $G^j \in \mathbf{R}^{n \times n}$ is a spline basis matrix with entries $G^j_{ii'} = g^j_{i'}(y^{(i)}_j)$, and $\Omega^j$ is the smoothing spline penalty matrix containing integrated products of pairs of twice differentiated basis functions. The MQGM problem in (4) can be translated into a finite-dimensional form, very similar to what we have done in (5), but we omit this for brevity.

**Example 3: RKHS model**    Consider taking $\mathcal{F}_{\ell k j} = \mathcal{H}_j$, a univariate reproducing kernel Hilbert space (RKHS), with kernel function $\kappa^j(\cdot, \cdot)$. The representer theorem allows us to express each function $f_{\ell k j} \in \mathcal{H}_j$ in terms of the representers of evaluation, *i.e.*, $f_{\ell k j}(x) = \sum_{i=1}^{n} (\theta_{\ell k j})_i \kappa^j(x, y^{(i)}_j)$, for a coefficient vector $\theta_{\ell k j} \in \mathbf{R}^n$. The work of [34], on high-dimensional additive RKHS modeling, suggests a choice of exponent $\omega = 1$, and sparsity and smoothness penalties

$$P_1(f_{\ell k j}) = \|K^j \theta_{\ell k j}\|_2 \quad \text{and} \quad P_2(f_{\ell k j}) = \sqrt{\theta^T_{\ell k j} K^j \theta_{\ell k j}},$$

respectively, where $K^j \in \mathbf{R}^{n \times n}$ is the kernel matrix with entries $K^j_{ii'} = \kappa^j(y^{(i)}_j, y^{(i')}_j)$. Again, the MQGM problem in (4) can be written in finite-dimensional form, now an SDP, omitted for brevity.

**Structural constraints**    Several structural constraints can be placed on top of the MQGM optimization problem in order to guide the estimated component functions to meet particular shape requirements. An important example are *non-crossing constraints* (commonplace in nonparametric, multiple quantile regression [18, 38]): here, we optimize (4) jointly over $\ell = 1, \ldots, r$, subject to

$$b_{\ell k} + \sum_{j \neq k} f_{\ell k j}(y^{(i)}_j) \leq b_{\ell' k} + \sum_{j \neq k} f_{\ell' k j}(y^{(i)}_j), \quad \text{for all } \alpha_\ell < \alpha_{\ell'}, \text{ and } i = 1, \ldots, n. \quad (6)$$

This ensures that the estimated quantiles obey the proper ordering, at the observations. For concreteness, we consider the implications for the basis regression model, in Example 1 (similar statements hold for the other two models). For each $\ell = 1, \ldots, r$, denote by $F_{\ell k}(b_{\ell k}, \theta_{\ell k})$ the criterion in (5). Introducing the non-crossing constraints requires coupling (5) over $\ell = 1, \ldots, r$, so that we now have the following optimization problems, for each target variable $k = 1, \ldots, d$:

$$\underset{B_k, \Theta_k}{\text{minimize}} \sum_{\ell=1}^{r} F_{\ell k}(b_{\ell k}, \theta_{\ell k}) \quad \text{subject to} \quad (\mathbf{1} B^T_k + \Phi \Theta_k) D^T \geq 0, \quad (7)$$

where we denote $B_k = (b_{1k}, \ldots, b_{rk}) \in \mathbf{R}^r$, $\Phi \in \mathbf{R}^{n \times dm}$ the basis matrix as before, $\Theta_k \in \mathbf{R}^{dm \times r}$ given by column-stacking $\theta_{\ell k} \in \mathbf{R}^{dm}$, $\ell = 1, \ldots, r$, and $D \in \mathbf{R}^{(r-1) \times r}$ is the usual discrete difference operator. (The inequality in (7) is to be interpreted componentwise.) Computationally, coupling the subproblems across $\ell = 1, \ldots, r$ clearly adds to the overall difficulty of the MQGM, but statistically this coupling acts as a regularizer, by constraining the parameter space in a useful way, thus increasing our efficiency in fitting multiple quantile levels from the given data.

For a triplet $\ell, k, j$, *monotonicity constraints* are also easy to add, *i.e.*, $f_{\ell k j}(y^{(i)}_j) \leq f_{\ell k j}(y^{(i')}_j)$ for all $y^{(i)}_j < y^{(i')}_j$. *Convexity constraints*, where we require $f_{\ell k j}$ to be convex over the observations, for a particular $\ell, k, j$, are also straightforward. Lastly, *strong non-crossing constraints*, where we enforce (6) over all $z \in \mathbf{R}^d$ (not just over the observations) are also possible with positive basis functions.

**Exogenous variables and conditional random fields**    So far, we have considered modeling the joint distribution $\mathbf{Pr}(y_1, \ldots, y_d)$, corresponding to learning a Markov random field (MRF). It is not hard to extend our framework to model the conditional distribution $\mathbf{Pr}(y_1, \ldots, y_d | x_1, \ldots, x_p)$ given some exogenous variables $x_1, \ldots, x_p$, corresponding to learning a conditional random field (CRF). To extend the basis regression model, we introduce the additional parameters $\theta^x_{\ell k} \in \mathbf{R}^p$ in (5), and the loss now becomes $\psi_{\alpha_\ell}(Y_k - b_{\ell k} \mathbf{1}^T - \Phi \theta_{\ell k} - X \theta^x_{\ell k})$, where $X \in \mathbf{R}^{n \times q}$ is filled with the exogenous observations $x^{(1)}, \ldots, x^{(n)} \in \mathbf{R}^q$; the other models are changed similarly.

## 4    Basic properties and theory

### 4.1    Quantiles and conditional independence

In the model (3), when a particular variable $y_j$ has no contribution, *i.e.*, satisfied $f^*_{\ell k j} = 0$ across all quantile levels $\alpha_\ell, \ell = 1, \ldots, r$, what does this imply about the conditional independence between $y_k$ and $y_j$, given the rest? Outside of the multivariate normal model (where the feature transformations need only be linear), nothing can be said in generality. But we argue that conditional independence can be understood in a certain *approximate sense* (*i.e.*, in a projected approximation of the data generating model). We begin with a simple lemma. Its proof is elementary, and given in the supplement.

**Lemma 4.1.** *Let $U, V, W$ be random variables, and suppose that all conditional quantiles of $U|V, W$ do not depend on $V$, i.e., $Q_{U|V,W}(\alpha) = Q_{U|W}(\alpha)$ for all $\alpha \in [0, 1]$. Then $U$ and $V$ are conditionally independent given $W$.*

By the lemma, if we knew that $Q_{U|V,W}(\alpha) = h(\alpha, U, W)$ for a function $h$, then it would follow that $U, V$ are conditionally independent given $W$ (*n.b.*, the converse is true, as well). The MQGM problem in (4), with sparsity imposed on the coefficients, essentially aims to achieve such a representation for the conditional quantiles; of course we cannot use a *fully nonparametric* representation of the conditional distribution $y_k|y_{\neg k}$ and instead we use an *r-step approximation* to the conditional cumulative distribution function (CDF) of $y_k|y_{\neg k}$ (corresponding to estimating $r$ conditional quantiles), and (say) in the basis regression model, limit the dependence on conditioning variables to be in terms of an additive function of RBFs in $y_j$, $j \neq k$. Thus, if at the solution in (5) we find that $\hat{\theta}_{\ell k j} = 0$, $\ell = 1, \ldots, r$, we may interpret this to mean that $y_k$ and $y_j$ are conditionally independent given the remaining variables, but according to the distribution defined by the *projection* of $y_k|y_{\neg k}$ onto the space of models considered in (5) (*r*-step conditional CDFs, which are additive expansions in $y_j$, $j \neq k$). This interpretation is no more tenuous (arguably, less so, as the model space here is much larger) than that needed when applying standard neighborhood selection to non-Gaussian data.

## 4.2 Gibbs sampling and the "joint" distribution

When specifying a form for the conditional distributions in a pseudolikelihood approximation as in (2), it is natural to ask: what is the corresponding joint distribution? Unfortunately, for a general collection of conditional distributions, there need not exist a compatible joint distribution, even when all conditionals are continuous [41]. Still, pseudolikelihood approximations (a special case of composite likelihood approximations), possess solid theoretical backing, in that maximizing the pseudolikelihood relates closely to minimizing a certain (expected composite) Kullback-Leibler divergence, measured to the true conditionals [39]. Recently, [7, 44] made nice progress in describing specific conditions on conditional distributions that give rise to a valid joint distribution, though their work was specific to exponential families. A practical answer to the question of this subsection is to use Gibbs sampling, which attempts to draw samples consistent with the fitted conditionals; this is precisely the observation of [13], who show that Gibbs sampling from discrete conditionals converges to a unique stationary distribution, although this distribution may not actually be compatible with the conditionals. The following result establishes the analogous claim for continuous conditionals; its proof is in the supplement. We demonstrate the practical value of Gibbs sampling through various examples in Section 6.

**Lemma 4.2.** *Assume that the conditional distributions $\mathbf{Pr}(y_k|y_{\neg k})$, $k = 1, \ldots, d$ take only positive values on their domain. Then, for any given ordering of the variables, Gibbs sampling converges to a unique stationary distribution that can be reached from any initial point. (This stationary distribution depends on the ordering.)*

## 4.3 Graph structure recovery

When $\log d = O(n^{2/21})$, and we assume somewhat standard regularity conditions (listed as A1–A4 in the supplement), the MQGM estimate recovers the underlying conditional independencies with high probability (interpreted in the projected model space, as explained in Section 4.1). Importantly, we do not require a Gaussian, sub-Gaussian, or even parametric assumption on the data generating process; instead, we assume i.i.d. draws $y^{(1)}, \ldots, y^{(n)} \in \mathbf{R}^d$, where the conditional distributions $y_k|y_{\neg k}$ have quantiles specified by the model in (3) for $k = 1, \ldots, d, \ell = 1, \ldots, r$, and further, each $f^*_{\ell k j}(x) = \theta^T_{\ell k j} \phi^j(x)^*$ for coefficients $\theta^*_{\ell k j} \in \mathbf{R}^m$, $j = 1, \ldots, d$, as in the basis expansion model.

Let $E^*$ denote the corresponding edge set of conditional dependencies from these neighborhood models, *i.e.*, $\{k, j\} \in E^* \iff \max_{\ell=1,\ldots,r} \max\{\|\theta^*_{\ell k j}\|_2, |\theta^*_{\ell j k}\|_2\} > 0$. We define the estimated edge set $\hat{E}$ in the analogous way, based on the solution in (5). Without a loss of generality, we assume the features have been scaled to satisfy $\|\Phi_j\| \leq \sqrt{n}$ for all $j = 1, \ldots, dm$. The following is our recovery result; its proof is provided in the supplement.

**Theorem 4.3.** *Assume $\log d = O(n^{2/21})$, and conditions A1–A4 in the supplement. Assume that the tuning parameters $\lambda_1, \lambda_2$ satisfy $\lambda_1 \asymp (mn \log(d^2 mr/\delta) \log^3 n)^{1/2}$ and $\lambda_2 = o(n^{41/42}/\theta^*_{\max})$, where $\theta^*_{\max} = \max_{\ell,k,j} \|\theta^*_{\ell k j}\|_2$. Then for $n$ sufficiently large, the MQGM estimate in (5) exactly recovers the underlying conditional dependencies, i.e., $\hat{E} = E^*$, with probability at least $1 - \delta$.*

The theorem shows that the nonzero pattern in the MQGM estimate identifies, with high probability, the underlying conditional independencies. But to be clear, we emphasize that the MQGM estimate is *not* an estimate of the inverse covariance matrix itself (this is also true of neighborhood regression, SpaceJam of [40], and many other methods for learning graphical models).

## 5  Computational approach

By design, the MQGM problem in (5) separates into $d$ subproblems, across $k = 1, \ldots, d$ (it therefore suffices to consider only a single subproblem, so we omit notational dependence on $k$ for auxiliary variables). While these subproblems are challenging for off-the-shelf solvers (even for only moderately-sized graphs), the key terms here all admit efficient *proximal operators* [32], which makes operator splitting methods like the alternating direction method of multipliers [5] a natural choice. As an illustration, we consider the non-crossing constraints in the basis regression model below. Reparameterizing our problem, so that we may apply ADMM, yields:

$$\begin{aligned}
&\underset{\Theta_k, B_k, V, W, Z}{\text{minimize}} \quad \psi_{\mathcal{A}}(Z) + \lambda_1 \sum_{\ell=1}^r \sum_{j=1}^d \|W_{\ell j}\|_2 + \tfrac{\lambda_2}{2}\|W\|_F^2 + I_+(VD^T) \\
&\text{subject to} \qquad V = \mathbf{1}B_k^T + \Phi\Theta_k, \; W = \Theta_k, \; Z = Y_k\mathbf{1}^T - \mathbf{1}B_k^T - \Phi\Theta_k,
\end{aligned} \tag{8}$$

where for brevity $\psi_{\mathcal{A}}(A) = \sum_{\ell=1}^r \sum_{j=1}^d \psi_{\alpha_\ell}(A_{\ell j})$, and $I_+(\cdot)$ is the indicator function of the space of elementwise nonnegative matrices. The augmented Lagrangian associated with (8) is:

$$L_\rho(\Theta_k, B_k, V, W, Z, U_V, U_W, U_Z) = \psi_{\mathcal{A}}(Z) + \lambda_1 \sum_{\ell=1}^r \sum_{j=1}^d \|W_{\ell j}\|_2 + \frac{\lambda_2}{2}\|W\|_F^2 + I_+(VD^T)$$

$$+ \frac{\rho}{2}\Big( \|\mathbf{1}B_k^T + \Phi\Theta_k - V + U_V\|_F^2 + \|\Theta_k - W + U_W\|_F^2 + \|Y_k\mathbf{1}^T - \mathbf{1}B_k^T - \Phi\Theta_k - Z + U_Z\|_F^2 \Big), \tag{9}$$

where $\rho > 0$ is the augmented Lagrangian parameter, and $U_V, U_W, U_Z$ are dual variables corresponding to the equality constraints on $V, W, Z$, respectively. Minimizing (9) over $V$ yields:

$$V \leftarrow P_{\text{iso}}\left(\mathbf{1}B_k^T + \Phi\Theta_k + U_V\right), \tag{10}$$

where $P_{\text{iso}}(\cdot)$ denotes the row-wise projection operator onto the isotonic cone (the space of componentwise nondecreasing vectors), an $O(nr)$ operation here [15]. Minimizing (9) over $W_{\ell j}$ yields the update:

$$W_{\ell j} \leftarrow \frac{(\Theta_k)_{\ell j} + (U_W)_{\ell j}}{1 + \lambda_2/\rho}\left(1 - \frac{\lambda_1/\rho}{\|(\Theta_k)_{\ell j} + (U_W)_{\ell j}\|_2}\right)_+, \tag{11}$$

where $(\cdot)_+$ is the positive part operator. This can be seen by deriving the proximal operator of the function $f(x) = \lambda_1\|x\|_2 + (\lambda_2/2)\|x\|_2^2$. Minimizing (9) over $Z$ yields the update:

$$Z \leftarrow \mathbf{prox}_{(1/\rho)\psi_{\mathcal{A}}}(Y_k\mathbf{1}^T - \mathbf{1}b_k^T - \Phi\Theta_k + U_Z), \tag{12}$$

where $\mathbf{prox}_f(\cdot)$ denotes the proximal operator of a function $f$. For the multiple quantile loss function $\psi_{\mathcal{A}}$, this is a kind of generalized soft-thresholding. The proof is given in the supplement.

**Lemma 5.1.** *Let $P_+(\cdot)$ and $P_-(\cdot)$ be the elementwise positive and negative part operators, respectively, and let $a = (\alpha_1, \ldots, \alpha_r)$. Then $\mathbf{prox}_{t\psi_{\mathcal{A}}}(A) = P_+(A - t\mathbf{1}a^T) + P_-(A - t\mathbf{1}a^T)$.*

Finally, differentiation in (9) with respect to $B_k$ and $\Theta_k$ yields the simultaneous updates:

$$\begin{bmatrix} \Theta_k \\ B_k^T \end{bmatrix} \leftarrow \frac{1}{2}\begin{bmatrix} \Phi^T\Phi + \frac{1}{2}I & \Phi^T\mathbf{1} \\ \mathbf{1}^T\Phi & \mathbf{1}^T\mathbf{1} \end{bmatrix}^{-1}\bigg( [I\ \mathbf{0}]^T(W - U_W) +$$

$$[\Phi\ \mathbf{1}]^T(Y_k\mathbf{1}^T - Z + U_Z + V - U_V)\bigg). \tag{13}$$

A complete description of our ADMM algorithm for solving the MQGM problem is in the supplement.

**Gibbs sampling**    Having fit the conditionals $y_k|y_{\neg k}$, $k = 1, \ldots d$, we may want to make predictions or extract joint distributions over subsets of variables. As discussed in Section 4.2, there is no general analytic form for these joint distributions, but the pseudolikelihood approximation underlying the MQGM suggests a natural Gibbs sampler. A careful implementation that respects the additive model in (3) yields a highly efficient Gibbs sampler, especially for CRFs; the supplement gives details.

# 6 Empirical examples

## 6.1 Synthetic data

We consider synthetic examples, comparing the MQGM to neighborhood selection (MB), the graphical lasso (GLasso), SpaceJam [40], the nonparanormal skeptic [26], TIGER [24], and neighborhood selection using the absolute loss (Laplace).

**Ring example** As a simple but telling example, we drew $n = 400$ samples from a "ring" distribution in $d = 4$ dimensions. Data were generated by drawing a random angle $\nu \sim \text{Uniform}(0, 1)$, a random radius $R \sim \mathcal{N}(0, 0.1)$, and then computing the coordinates $y_1 = R \cos \nu$, $y_2 = R \sin \nu$ and $y_3, y_4 \sim \mathcal{N}(0, 1)$, *i.e.*, $y_1$ and $y_2$ are the only dependent variables here. The MQGM was used with $m = 10$ basis functions (RBFs), and $r = 20$ quantile levels. The left panel of Figure 1 plots samples (blue) of the coordinates $y_1, y_2$ as well as new samples from the MQGM (red) fitted to these same (blue) samples, obtained by using our Gibbs sampler; the samples from the MQGM appear to closely match the samples from the underlying ring. The main panel of Figure 1 shows the conditional dependencies recovered by the MQGM, SpaceJam, GLasso, and MB (plots for the other methods are given in the supplement), when run on the ring data. We visualize these dependencies by forming a $d \times d$ matrix with the cell $(j, k)$ set to black if $j, k$ are conditionally dependent given the others, and white otherwise. Across a range of tuning parameters for each method, the MQGM is the only one that successfully recovers the underlying conditional dependencies, at some point along its solution path. In the supplement, we present an evaluation of the conditional CDFs given by each method, when run on the ring data; again, the MQGM performs best in this setting.

**Larger examples** To investigate performance at larger scales, we drew $n \in \{50, 100, 300\}$ samples from a multivariate normal and Student $t$-distribution (with 3 degrees of freedom), both in $d = 100$ dimensions, both parameterized by a random, sparse, diagonally dominant $d \times d$ inverse covariance matrix, following the procedure in [33, 17, 31, 1]. Over the same set of sample sizes, with $d = 100$, we also considered an autoregressive setup in which we drew samples of pairs of adjacent variables from the ring distribution. In all three data settings (normal, $t$, and autoregressive), we used $m = 10$ and $r = 20$ for the MQGM. To summarize the performances, we considered a range of tuning parameters for each method, computed corresponding false and true positive rates (in detecting conditional dependencies), and then computed the corresponding area under the curve (AUC), following, *e.g.*, [33, 17, 31, 1]. Table 1 reports the median AUCs (across 50 trials) for all three of these examples; the MQGM outperforms all other methods on the autoregressive example; on the normal and Student $t$ examples, it performs quite competitively.

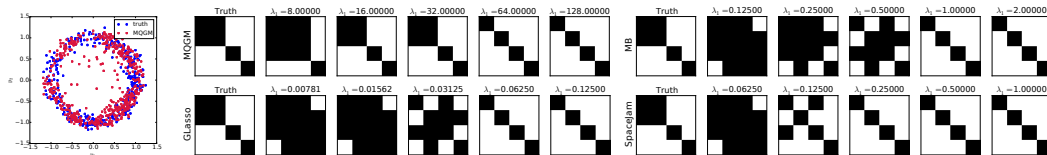

**Figure 1:** Left: data from the ring distribution (blue) as well as new samples from the MQGM (red) fitted to the same (blue) data, obtained by using our Gibbs sampler. Right: conditional dependencies recovered by the MQGM, MB, GLasso, and SpaceJam on the ring data; black means conditional dependence. The MQGM is the only method that successfully recovers the underlying conditional dependencies along its solution path.

**Table 1:** AUC values for the MQGM, MB, GLasso, SpaceJam, the nonparanormal skeptic, TIGER, and Laplace for the normal, $t$, and autoregressive data settings; higher is better, best in **bold**.

|  | Normal | | | Student $t$ | | | Autoregressive | | |
|---|---|---|---|---|---|---|---|---|---|
|  | $n = 50$ | $n = 100$ | $n = 300$ | $n = 50$ | $n = 100$ | $n = 300$ | $n = 50$ | $n = 100$ | $n = 300$ |
| MQGM | **0.953** | **0.976** | 0.988 | **0.928** | 0.947 | 0.981 | **0.726** | **0.754** | **0.955** |
| MB | 0.850 | 0.959 | 0.994 | 0.844 | 0.923 | 0.988 | 0.532 | 0.563 | 0.725 |
| GLasso | 0.908 | 0.964 | **0.998** | 0.691 | 0.605 | 0.965 | 0.541 | 0.620 | 0.711 |
| SpaceJam | 0.889 | 0.968 | 0.997 | 0.893 | **0.965** | 0.993 | 0.624 | 0.708 | 0.854 |
| Nonpara. | 0.881 | 0.962 | 0.996 | 0.862 | 0.942 | **0.998** | 0.545 | 0.590 | 0.612 |
| TIGER | 0.732 | 0.921 | 0.996 | 0.420 | 0.873 | 0.989 | 0.503 | 0.518 | 0.718 |
| Laplace | 0.803 | 0.931 | 0.989 | 0.800 | 0.876 | 0.991 | 0.530 | 0.554 | 0.758 |

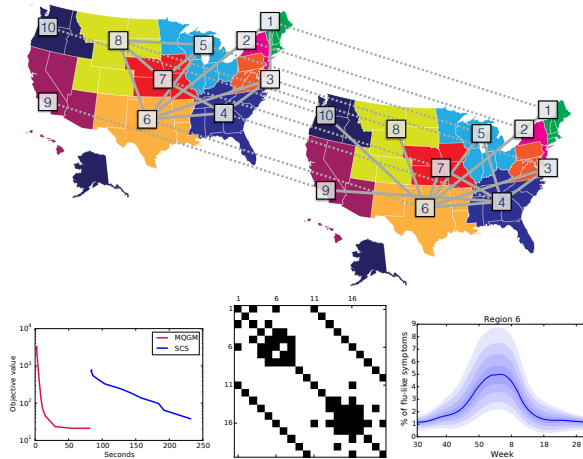

**Figure 2:** Top panel and bottom row, middle panel: conditional dependencies recovered by the MQGM on the flu data; each of the first ten cells corresponds to a region of the U.S., and black means dependence. Bottom row, left panel: wallclock time (in seconds) for solving one subproblem using ADMM versus SCS. Bottom row, right panel: samples from the fitted marginal distribution of the weekly flu incidence rates at region 6; samples at larger quantiles are shaded lighter, and the median is in darker blue.

## 6.2 Modeling flu epidemics

We study $n = 937$ weekly flu incidence reports from September 28, 1997 through August 30, 2015, across 10 regions in the United States (see the top panel of Figure 2), obtained from [6]. We considered $d = 20$ variables: the first 10 encode the current week's flu incidence (precisely, the percentage of doctor's visits in which flu-like symptoms are presented) in the 10 regions, and the last 10 encode the same but for the prior week. We set $m = 5$, $r = 99$, and also introduced exogenous variables to encode the week numbers, so $p = 1$. Thus, learning the MQGM here corresponds to learning the structure of a spatiotemporal graphical model, and reduces to solving 20 multiple quantile regression subproblems, each of dimension $(19 \times 5 + 1) \times 99 = 9504$. All subproblems took about 1 minute on a 6 core 3.3 Ghz Core i7 X980 processor.

The bottom left panel in Figure 2 plots the time (in seconds) taken for solving one subproblem using ADMM versus SCS [30], a cone solver that has been advocated as a reasonable choice for a class of problems encapsulating (4); ADMM outperforms SCS by roughly two orders of magnitude. The bottom middle panel of Figure 2 presents the conditional independencies recovered by the MQGM. Nonzero entries in the upper left $10 \times 10$ submatrix correspond to dependencies between the $y_k$ variables for $k = 1, \ldots, 10$; *e.g.*, the nonzero (0,2) entry suggests that region 1 and 3's flu reports are dependent. The lower right $10 \times 10$ submatrix corresponds to the $y_k$ variables for $k = 11, \ldots, 20$, and the nonzero banded entries suggest that at any region the previous week's flu incidence (naturally) influences the next week's. The top panel of Figure 2 visualizes these relationships by drawing an edge between dependent regions; region 6 is highly connected, suggesting that it may be a bellwether for other regions, roughly in keeping with the current understanding of flu dynamics. To draw samples from the fitted distributions, we ran our Gibbs sampler over the year, generating 1000 total samples, making 5 passes over all coordinates between each sample, and with a burn-in period of 100 iterations. The bottom right panel of Figure 2 plots samples from the marginal distribution of the percentages of flu reports at region 6 (other regions are in the supplement) throughout the year, revealing the heteroskedastic nature of the data.

For space reasons, our last example, on wind power data, is presented in the supplement.

## 7 Discussion

We proposed and studied the Multiple Quantile Graphical Model (MQGM). We established theoretical and empirical backing to the claim that the MQGM is capable of compactly representing relationships between heteroskedastic non-Gaussian variables. We also developed efficient algorithms for both estimation and sampling in the MQGM. All in all, we believe that our work represents a step forward in the design of flexible yet tractable graphical models.

**Acknowledgements**     AA was supported by DOE Computational Science Graduate Fellowship DE-FG02-97ER25308. JZK was supported by an NSF Expeditions in Computation Award, CompSustNet, CCF-1522054. RJT was supported by NSF Grants DMS-1309174 and DMS-1554123.

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
