[Supplementary Material]

# Supplement to "The Multiple Quantile Graphical Model"

**Alnur Ali**
Machine Learning Department
Carnegie Mellon University
alnurali@cmu.edu

**J. Zico Kolter**
Computer Science Department
Carnegie Mellon University
zkolter@cs.cmu.edu

**Ryan J. Tibshirani**
Department of Statistics
Carnegie Mellon University
ryantibs@cmu.edu

This document contains proofs and supplementary details for the paper "The Multiple Quantile Graphical Model". All section, equation, and figure numbers in this supplementary document are preceded by the letter S (all numbering without an S refers to the main paper).

## S.1 Proof of Lemma 4.1

If the conditional quantiles satisfy $Q_{U|V,W}(\alpha) = Q_{U|W}(\alpha)$ for all $\alpha \in [0,1]$, then the conditional CDF must obey the same property, *i.e.*, $F_{U|V,W}(t) = F_{U|W}(t)$ for all $t$ in the support of $U$. This is simply because any CDF may be expressed in terms of its corresponding quantile function (*i.e.*, inverse CDF), as in

$$F_{U|V,W}(t) = \sup\{\alpha \in [0,1] : Q_{U|V,W}(\alpha) \le t\},$$

and the right-hand side does not depend on $V$, so neither can the left-hand side. But this precisely implies that the distribution of $U|V,W$ equals that of $U|W$, *i.e.*, $U$ and $V$ are conditionally independent given $W$. We note that the converse of the statement in the lemma is true as well, by just reversing all the arguments here. □

## S.2 Proof of Lemma 4.2

This result can be seen as a generalization of Theorem 3 in [5].

First, we define an *iteration* of Gibbs sampling to be a single pass through all the variables (without a loss of generality, we take this order to be $y_1, \ldots, y_d$). Now, consider a particular iteration of Gibbs sampling; let $\tilde{y}_1, \ldots, \tilde{y}_d$ be the values assigned to the variables on the previous iteration. Then the transition kernel for our Gibbs sampler is given by

$$\mathbf{Pr}(y_1, \ldots, y_d | \tilde{y}_1, \ldots, \tilde{y}_d) = \mathbf{Pr}(y_d | y_{d-1}, \ldots, y_1, \tilde{y}_1, \ldots, \tilde{y}_d)\, \mathbf{Pr}(y_{d-1}, \ldots, y_1 | \tilde{y}_1, \ldots, \tilde{y}_d) \quad \text{(S.1)}$$
$$= \mathbf{Pr}(y_d | y_{d-1}, \ldots, y_1)\, \mathbf{Pr}(y_{d-1}, \ldots, y_1 | \tilde{y}_1, \ldots, \tilde{y}_d) \quad \text{(S.2)}$$
$$= \mathbf{Pr}(y_d | y_{d-1}, \ldots, y_1)\, \mathbf{Pr}(y_{d-1} | y_{d-2}, \ldots y_1, \tilde{y}_d) \cdots \mathbf{Pr}(y_1 | \tilde{y}_2, \ldots, \tilde{y}_d), \quad \text{(S.3)}$$

where (S.1) follows by the definition of conditional probability, (S.2) by conditional independence, and (S.3) by repeated applications of these tools. Since each conditional distribution is assumed to be (strictly) positive, we have that the transition kernel is also positive, which in turn implies [2, page 544] that the induced Markov chain is ergodic with a unique stationary distribution that can be reached from any initial point. □

## S.3 Statement and discussion of regularity conditions for Theorem 4.3

For each $k = 1, \ldots, r$, $\ell = 1, \ldots, r$, let us define the "effective" (independent) error terms $\epsilon_{\ell k i} = y_k^{(i)} - b_{\ell k}^* - \sum_{j \ne k} \phi(y_j^{(i)})^T \theta_{\ell k j}^*$, over $i = 1, \ldots, n$. Denote by $F_{\epsilon_{\ell k}}$ the conditional CDF of

$\epsilon_{\ell k i}|y_{\neg k}^{(i)}$, $i = 1, \ldots, n$, which by construction satisfies $F_{\epsilon_{\ell k}}(0) = \alpha_\ell$. Also define the underlying support

$$S_{\ell k} = \left\{ j \in \{1, \ldots, d\} : \theta_{\ell k j}^* \neq 0 \right\}.$$

Here we take a moment to explain a somewhat subtle indexing issue with the columns of the feature matrix $\Phi \in \mathbf{R}^{n \times dm}$. For a single fixed index $j = 1, \ldots, d$, we will extract an appropriate block of columns of $\Phi \in \mathbf{R}^{n \times dm}$, corresponding to the basis expansion of variable $j$, by writing $\Phi_j$. More precisely, we use $\Phi_j$ to denote the block of $m$ columns

$$[\Phi_{(j-1)m+1}, \Phi_{(j-1)m+2}, \ldots, \Phi_{jm}]. \tag{S.4}$$

We do this because it simplifies notation considerably. (Occasionally, to be transparent, we will use the more exhaustive notation on the right-hand side in (S.4), but this is to be treated as an exception, and the default is to use the concise notation as in $\Phi_j$.) The same rule will be used for subsets of indices among $1, \ldots, d$, so that $\Phi_{S_{\ell k}}$ denotes the appropriate block of $m|S_{\ell k}|$ columns corresponding to the basis expansions of the variables in $S_{\ell k}$.

For all $k = 1, \ldots, d$, $\ell = 1, \ldots, r$, we will assume the following regularity conditions.

**A1.** *Groupwise irrepresentability:* for $j \in S_{\ell k}^c$, we require that $\|\Phi_j^T \Phi_{S_{\ell k}}\|_F < \lambda_1 / (6 f_{\epsilon_{\ell k}}(0)\gamma)$, where $S_{\ell k} = \{j \in \{1, \ldots, dm\} : \theta_{\ell k j}^* \neq 0\}$, $f_{\epsilon_{\ell k}}$ is the density of $F_{\epsilon_{\ell k}}$, and $\gamma > 0$ is a quantity prescribed by Lemma S.5.

**A2.** *Distributional smoothness:* we assume that $|F_{\epsilon_{\ell k}}(x) - F_{\epsilon_{\ell k}}(0) - x f_{\epsilon_{\ell k}}(0)| \leq C_1 x^2$ for all $|x| \leq C_2$, where $C_1, C_2 > 0$ are constants.

**A3.** *Correlation restriction:* we assume that $C_3 \leq (f_{\epsilon_{\ell k}}(0)/n)\lambda_{\min}(\Phi_{S_{\ell k}}^T \Phi_{S_{\ell k}}) \leq C_4$ for constants $C_3, C_4 > 0$, where $\lambda_{\min}(A)$ denotes the minimum eigenvalue of $A$.

**A4.** *Basis and support size restrictions:* we assume that $m = O(n^{1/9})$ and $s = O(n^{1/21})$, where $s = |S_{\ell k}|$. We also assume, with probability tending to one, that $\Phi_{\max} = \Omega(1)$ and $\Phi_{\max} = o(n^{1/21}/\log^{1/2} n)$, where we write $\Phi_{\max}$ to denote the maximum absolute entry of the basis matrix $\Phi$.

Next, we provide some intuition for these conditions.

*Condition A1.* Fix some $j \in S_{\ell k}^c$. For notational convenience, we let

$$A = \Phi_j^T \Phi_{S_{\ell k}} \in \mathbf{R}^{m \times sm}.$$

Observe that each entry of $A$ can be expressed as

$$A_{ip} = n\rho_{i,p}\|\Phi_{(j-1)m+i}\|_2\|\Phi_p\|_2, \tag{S.5}$$

for $i = 1, \ldots, m$, $p$ denoting an index into the basis expansion of the columns $\Phi_{S_{\ell k}}$, and $\rho_{i,p}$ denoting the sample correlation coefficient for the columns $\Phi_i$ and $\Phi_p$. Since $\|A_p\|_F \leq \sqrt{m}\|A_p\|_\infty$, we have that

$$\max_{i,p} \rho_{i,p} < \frac{\lambda_1}{6n^2 f_{\epsilon_{\ell k}}(0)\sqrt{m}}$$

is sufficient for condition A1; here, we have also used the column scaling assumption $\|\Phi_p\|_2 \leq \sqrt{n}$.

So, roughly speaking, bounded correlation between each pair of columns in the submatrices $\Phi_j$ and $\Phi_{S_{\ell k}}$ is enough for condition A1 to hold; note that this is trivially satisfied when $\Phi_i^T \Phi_p = 0$, for $i = 1, \ldots, m$, and $p$ as defined above. Condition A1 is therefore similar to, *e.g.*, the mutual incoherence condition of [7] for the lasso, which is given by

$$\left\|\Phi_{S^c}^T \Phi_S \left(\Phi_S^T \Phi_S\right)^{-1}\right\|_\infty \leq 1 - \tilde{\gamma} \iff \max_{j \in S^c} \left\|\left(\Phi_S^T \Phi_S\right)^{-1} \Phi_S^T \Phi_j\right\|_1 \leq 1 - \tilde{\gamma},$$

where again $\Phi_S$ extracts the appropriate block of columns of $\Phi$, $\|\cdot\|_\infty$ here denotes the $\ell_\infty$ operator norm (maximum $\ell_1$ norm of a row), $\|\cdot\|_1$ here denotes the elementwise $\ell_1$ norm, and $\tilde{\gamma} \in (0, 1]$ is a constant. This condition can be seen as requiring bounded correlation between each column in the submatrix $\Phi_{S^c}$ and all columns in the submatrix $\Phi_S$.

*Condition A2.* This condition is similar to requiring that $f_{\epsilon\ell k}(x)$ be Lipschitz, over some $x$ in a neighborhood of 0. We can show that the Laplace distribution, *e.g.*, satisfies this condition.

The density and distribution functions for the Laplace distribution with location zero and unit scale are given by

$$f_{\epsilon\ell k}(x) = (1/2)\exp(-|x|)$$

and

$$F_{\epsilon\ell k}(x) = \begin{cases} 1 - (1/2)\exp(-x) & \text{if } x \geq 0 \\ (1/2)\exp(x) & \text{if } x < 0, \end{cases}$$

respectively.

Now, suppose $0 \leq x \leq C_2$. Then we can express condition A2 as

$$|f_{\epsilon\ell k}(x) - f_{\epsilon\ell k}(0) - xf_{\epsilon\ell k}(0)| \leq C_1 x^2 \iff -2C_1 x^2 \leq \exp(-x) + x - 1 \leq 2C_1 x^2.$$

For the first inequality, since $1 - x \leq \exp(-x)$, it is sufficient to check that $0 \leq C_1 x^2$, which is true for $C_1 > 0$ and all $x$. For the second inequality, by differentiating and again using $1 - x \leq \exp(-x)$, we have that the function

$$2C_1 x^2 - \exp(-x) - x + 1 \tag{S.6}$$

is nondecreasing in $x \geq 0$; thus, it is sufficient to check that this function is nonnegative for $x = 0$, which is true.

Now, suppose $-C_2 \leq x < 0$. Then we can express condition A2 as

$$|f_{\epsilon\ell k}(x) - f_{\epsilon\ell k}(0) - xf_{\epsilon\ell k}(0)| \leq C_1 x^2 \iff -2C_1 x^2 \leq \exp(x) - x - 1 \leq 2C_1 x^2.$$

By symmetry with the preceding case, the first inequality here holds. The second inequality here also holds, since $\exp(x) - 2C_1 x^2 - x - 1$ is continuous and increasing in $x < 0$; taking the limit as $x \uparrow 0$ gives that this function is nonpositive as required.

*Condition A3.* This condition is a generalization of the minimum eigenvalue condition of [7], *i.e.*, $c_{\min} \leq \lambda_{\min}\left((1/n)\Phi_S^T\Phi_S\right)$, for some constant $c_{\min} > 0$, and where we write $\Phi_S$ to extract the appropriate block of columns of $\Phi$.

*Condition A4.* This condition allows the number of basis functions $m$ in the expansion to grow with $n$, at a polynomial rate (with fractional exponent). This is roughly in line with standard nonparametric regression; *e.g.*, when estimating a continuous differentiable function via a spline expansion, one typically takes the number of basis functions $m$ to scale as $n^{1/3}$ [4]. The condition also restricts, for any given variable, the number of variables $s$ that contribute to its neighborhood model to be polynomial in $n$ (with a smaller fractional exponent).

Finally, the condition assumes that the entries of the basis matrix $\Phi$ (*i.e.*, the matrix of transformed variables) to be at least of constant order, and at most of polynomial order (with small fractional exponent), with $n$. We note that this implicitly places a restriction on the tails of distribution governing the data $y_j^{(i)}$, $i = 1, \ldots, n$, $j = 1, \ldots, d$. However, the restriction is not a strong one, because it allows the maximum to grow polynomially large with $n$ (whereas a logarithmic growth would be expected, *e.g.*, for normal data). Furthermore, it is possible to trade off the restrictions on $m$, $s$, $\Phi_{\max}$, and $d$ (presented in the statement of the theorem), making each of these restrictions more or less stringent, if required.

## S.4 Proof of Theorem 4.3

The general strategy that we use here for support recovery is inspired by that in [3], for $\ell_1$-penalized quantile regression.

Fix some $k = 1, \ldots, d$ and $\ell = 1, \ldots, r$. We consider the conditional distribution $y_k|y_{\neg k}$, whose $\alpha_\ell$-quantile is assumed to satisfy (3). Hence, to be perfectly clear, all expectations and probability statements in what follows are to be interpreted with respect to the observations $y_k^{(i)}$, $i = 1, \ldots, n$ conditional on $y_j^{(i)}$, $i = 1, \ldots, n$, for $j \neq k$ (and thus we can treat the feature matrix $\Phi$ as fixed

throughout). In the setting assumed by the theorem, the conditional quantile model in (3) is, more explicitly,

$$Q_{y_k|y_{\neg k}}(\alpha_\ell) = b_{\ell k}^* + \sum_{j \neq k}^{d} (\theta_{\ell k j}^*)^T \phi^j(y_j),$$

for some unknown parameters $b_{\ell k}^*$ and $\theta_{\ell k j}^*$, $j = 1, \ldots, d$. For simplicity, in this proof, we will drop the intercept term completely both from the model (denoted $b_{\ell k}^*$) and the optimization problem in (4) (here denoted $b_{\ell k}$) that defines the estimator in question. Including the intercept is not at all difficult, and it just requires some extra bookkeeping at various places. Recall that we define

$$S_{\ell k} = \left\{ j \in \{1, \ldots, d\} : \theta_{\ell k j}^* \neq 0 \right\},$$

and analogously define

$$\hat{S}_{\ell k} = \left\{ j \in \{1, \ldots, d\} : \hat{\theta}_{\ell k j} \neq 0 \right\},$$

where $\hat{\theta}_{\ell k} = (\hat{\theta}_{\ell k 1}, \ldots, \hat{\theta}_{\ell k d}) \in \mathbf{R}^{dm}$ is the solution in (5).

We will show that, with probability at least $1 - \delta/(dr)$, it holds that $S_{\ell k} = \hat{S}_{\ell k}$. A union bound (over all choices $k = 1, \ldots, d$ and $\ell = 1, \ldots, r$) will then tell us that $E^* = \hat{E}$ with probability at least $1 - \delta$, completing the proof.

To certify that $S_{\ell k} = \hat{S}_{\ell k}$, we will show that the unique solution in (5) is given by

$$\hat{\theta}_{\ell k(S_{\ell k})} = \tilde{\theta}_{\ell k(S_{\ell k})}, \quad \hat{\theta}_{\ell k(S_{\ell k}^c)} = 0, \tag{S.7}$$

where $\tilde{\theta}_{\ell k(S_{\ell k})}$ solves the "restricted" optimization problem:

$$\underset{\theta_{\ell k(S_{\ell k})}}{\text{minimize}} \ \psi_{\alpha_\ell}\left(Y_k - \Phi_{S_{\ell k}} \theta_{\ell k(S_{\ell k})}\right) + \lambda_1 \sum_{j \in S_{\ell k}} \|\theta_{\ell k j}\|_2 + \frac{\lambda_2}{2} \|\theta_{\ell k(S_{\ell k})}\|_2^2. \tag{S.8}$$

Now, to prove that $\hat{\theta}_{\ell k}$ as defined above in (S.7) indeed the solution in (5), we need to check that it satisfies the KKT conditions for (5), namely

$$\Phi_{S_{\ell k}}^T v_\ell \left(Y_k - \Phi_{S_{\ell k}} \tilde{\theta}_{\ell k(S_{\ell k})}\right) - \lambda_2 \tilde{\theta}_{\ell k(S_{\ell k})} = \lambda_1 u_{\ell k(S_{\ell k})}, \tag{S.9}$$

$$\Phi_{S_{\ell k}^c}^T v_\ell \left(Y_k - \Phi_{S_{\ell k}} \tilde{\theta}_{\ell k(S_{\ell k})}\right) = \lambda_1 u_{\ell k(S_{\ell k}^c)}, \tag{S.10}$$

where $v_\ell(Y_k - \Phi_{S_{\ell k}} \tilde{\theta}_{\ell k(S_{\ell k})}) \in \mathbf{R}^n$ is a subgradient of $\psi_{\alpha_\ell}(\cdot)$ at $Y_k - \Phi_{S_{\ell k}} \tilde{\theta}_{\ell k(S_{\ell k})}$, i.e.,

$$\left[ v_\ell \left(Y_k - \Phi_{S_{\ell k}} \tilde{\theta}_{\ell k(S_{\ell k})}\right) \right]_i = \alpha_\ell - I_-\left( y_k^{(i)} - \Phi_{i(S_{\ell k})} \tilde{\theta}_{\ell k(S_{\ell k})}\right), \quad i = 1, \ldots, n$$

where $I_-(\cdot)$ is the indicator function of the nonpositive real line, and where each $u_{\ell k j} \in \mathbf{R}^m$ is a subgradient of $\| \cdot \|_2$ at $\tilde{\theta}_{\ell k j}$, i.e.,

$$u_{\ell k j} \in \begin{cases} \{\tilde{\theta}_{\ell k j} / \|\tilde{\theta}_{\ell k j}\|_2\} & \text{if } \theta_{\ell k j} \neq 0 \\ \{x \in \mathbf{R}^m : \|x\|_2 \leq 1\} & \text{if } \theta_{\ell k j} = 0, \end{cases}$$

for $j = 1, \ldots, d$. Note that, since $\tilde{\theta}_{\ell k(S_{\ell k})}$ is optimal for the restricted problem (S.8), we know that there exists a collection of subgradients $u_{\ell k(S_{\ell k})}$ to satisfy (S.9), from the KKT conditions for (S.8) itself.

It remains to satisfy (S.10), and for this, we can use $u_{\ell k j} = \Phi_j^T v_\ell(Y_k - \Phi_{S_{\ell k}} \tilde{\theta}_{\ell k(S_{\ell k})})$ as a valid choice of subgradient, for each $j \in S_{\ell k}^c$, provided that

$$\left\| \Phi_j^T v_\ell \left(Y_k - \Phi_{S_{\ell k}} \tilde{\theta}_{\ell k(S_{\ell k})}\right) \right\|_2 < \lambda_1, \quad \text{for } j \in S_{\ell k}^c. \tag{S.11}$$

Define $z_j(\vartheta) = \Phi_j^T v_\ell(Y_k - \Phi_{S_{\ell k}} \vartheta)$, for $j \in S_{\ell k}^c$, and define a ball

$$B^* = \left\{ \vartheta \in \mathbf{R}^{sm} : \|\vartheta - \theta_{\ell k(S_{\ell k})}^*\|_2 \leq \gamma \right\},$$

where we write $s = |S_{\ell k}|$. To show (S.11), then, it suffices to show that

$$\underbrace{\tilde{\theta}_{\ell k(S_{\ell k})} \in B^*}_{E_1}, \quad \text{and} \quad \underbrace{\max_{j \in S_{\ell k}^c} \sup_{\vartheta \in B^*} \|z_j(\vartheta)\|_2 < \lambda_1}_{E_2}. \tag{S.12}$$

In Lemma S.5.1, given in Section S.5, it is shown that the event $E_1$ defined above occurs with probability at least $1 - \delta/(2dr)$, with a choice of radius

$$\gamma = C \left( \frac{\lambda_1 s \sqrt{m}}{n} + \sqrt{\frac{s \log n}{n}} \right),$$

for a constant $C > 0$. Below we show that $E_2$ occurs with probability at least $1 - \delta/(2dr)$, as well. For $j = 1, \ldots, d$, let us expand

$$z_j(\vartheta) = \underbrace{\Phi_j^T v_\ell(\epsilon_{\ell k})}_{\Delta_1^j} + \underbrace{\Phi_j^T \mathbf{E} \left[ v_\ell \left( Y_k - \Phi_{S_{\ell k}} \vartheta \right) - v_\ell(\epsilon_{\ell k}) \right]}_{\Delta_2^j} + $$

$$\underbrace{\Phi_j^T \left( v_\ell \left( Y_k - \Phi_{S_{\ell k}} \vartheta \right) - v_\ell(\epsilon_{\ell k}) - \mathbf{E} \left[ v_\ell(Y_k - \Phi_{S_{\ell k}} \vartheta) - v_\ell(\epsilon_{\ell k}) \right] \right)}_{\Delta_3^j}, \quad \text{(S.13)}$$

where $\epsilon_{\ell k} = (\epsilon_{\ell k 1}, \ldots, \epsilon_{\ell k n}) \in \mathbf{R}^n$ is a vector of the effective error terms, which recall, is defined by $\epsilon_{\ell k} = Y_k - \Phi \theta_{\ell k}^*$. Therefore, to show that the event $E_2$ in (S.12) holds, we can show that for each $p = 1, 2, 3$,

$$\max_{j \in S_{\ell k}^c} \sup_{\vartheta \in B^*} \|\Delta_p^j\|_2 < \frac{\lambda_1}{3}.$$

Further, to show that $E_2$ holds with probability at least $1 - \delta/(2dr)$, we can show that the above holds for $p = 1, 3$ each with probability at least $1 - \delta/(4dr)$, as the statement for $p = 2$ is deterministic. We now bound the terms $\Delta_1^j, \Delta_2^j, \Delta_3^j$ one by one.

*Bounding* $\|\Delta_1^j\|_2$. Fix $j \in S_{\ell k}^c$, and write

$$\Phi_j^T v_\ell(\epsilon_{\ell k}) = \left( \sum_{i=1}^n \Phi_{i,(j-1)m+1} v_\ell(\epsilon_{\ell k i}), \ldots, \sum_{i=1}^n \Phi_{i,jm} v_\ell(\epsilon_{\ell k i}) \right),$$

where, as a reminder that the above quantity is a vector, we have returned momentarily to the more exhaustive notation for indexing the columns of $\Phi$, as in the right-hand side of (S.4).

Straightforward calculations reveal that, for each $i = 1, \ldots, n$, and $p = 1, \ldots, m$,

$$\mathbf{E}\, \Phi_{i,(j-1)m+p} v_\ell(\epsilon_{\ell k i}) = 0, \quad \text{and} \quad -|\Phi_{i,(j-1)m+p}| \le \Phi_{i,(j-1)m+p} v_\ell(\epsilon_{\ell k i}) \le |\Phi_{i,(j-1)m+p}|.$$

Hence,

$$\mathbf{Pr} \left( \|\Phi_j^T v_\ell(\epsilon_{\ell k i})\|_2 \ge \sqrt{m} t \right) \le \mathbf{Pr} \left( \left| \sum_{i=1}^n \Phi_{i,(j-1)m+p} v_\ell(\epsilon_{\ell k i}) \right| \ge t, \text{ some } p = 1, \ldots, m \right)$$

$$\le \sum_{p=1}^m 2 \exp \left( - \frac{t^2}{2 \sum_{i=1}^n \Phi_{i,(j-1)m+p}^2} \right)$$

$$\le 2m \exp \left( - \frac{t^2}{2n} \right).$$

Above, the first inequality used the simple fact that $\|x\|_2 \le \sqrt{m}\|x\|_\infty$ for $x \in \mathbf{R}^m$; the second used Hoeffding's bound and the union bound; and the third used our assumption that the columns of $\Phi$ have norm at most $\sqrt{n}$. Therefore, taking $t = \lambda_1/(3\sqrt{m})$, we see that, by the above and the union bound,

$$\mathbf{Pr} \left( \max_{j \in S_{\ell k}^c} \|\Delta_1^j\|_2 < \frac{\lambda_1}{3} \right) \ge 1 - 2dm \exp \left( - \frac{\lambda_1^2}{18mn} \right).$$

By choosing $\lambda_1 = C' \sqrt{18mn \log(8d^2mr/\delta)}$ for a constant $C' > 0$, we see that the probability in question is at least $1 - \delta/(4dr)$, as desired.

*Bounding* $\|\Delta_2^j\|_2$. Recall that $F_{\epsilon_{\ell k}}(\cdot)$ is used to denote the CDF of the effective error distribution, and $f_{\epsilon_{\ell k}}(\cdot)$ is used for its its density. By construction, $F_{\epsilon_{\ell k}}(0) = \alpha_\ell$. Direct calculation, using the definition of $v_\ell(\cdot)$, shows that, for any $\vartheta \in B^*$, and each $i = 1, \ldots, n$,

$$\mathbf{E}\left[v_\ell(\epsilon_{\ell k}) - v_\ell\left(Y_k - \Phi_{S_{\ell k}}\vartheta\right)\right] = F_{\epsilon_{\ell k}}\left(\Phi_{S_{\ell k}}\left(\vartheta - \theta^*_{\ell k(S_{\ell k})}\right)\right) - F_{\epsilon_{\ell k}}(\mathbf{0}),$$

where we apply $F_{\epsilon_{\ell k}}$ componentwise, and so

$$\Phi_j^T \mathbf{E}\left[v_\ell(\epsilon_{\ell k}) - v_\ell\left(Y_k - \Phi_{S_{\ell k}}\vartheta\right)\right] = f_{\epsilon_{\ell k}}(0)\Phi_j^T\Phi_{S_{\ell k}}\left(\vartheta - \theta^*_{\ell k(S_{\ell k})}\right) + \Delta_4^j$$

with $\Delta_4^j \in \mathbf{R}^m$ being the appropriate remainder term, *i.e.*,

$$\left[\Delta_4^j\right]_t = \sum_{i=1}^n \Phi_{it}\left[F_{\epsilon_{\ell k}}\left(\Phi_{i(S_{\ell k})}\left(\vartheta - \theta^*_{\ell k(S_{\ell k})}\right)\right) - F_{\epsilon_{\ell k}}(0) - f_{\epsilon_{\ell k}}(0)\Phi_{i(S_{\ell k})}\left(\vartheta - \theta^*_{\ell k(S_{\ell k})}\right)\right],$$

for $t = j(m-1) + 1, \ldots, jm$.

Now, we have that

$$\left\|f_{\epsilon_{\ell k}}(0)\Phi_j^T\Phi_{S_{\ell k}}\left(\vartheta - \theta^*_{\ell k(S_{\ell k})}\right)\right\|_2 \leq f_{\epsilon_{\ell k}}(0)\left\|\Phi_j^T\Phi_{S_{\ell k}}\right\|_F\left\|\vartheta - \theta^*_{\ell k(S_{\ell k})}\right\|_2 \leq \frac{\lambda_1}{6},$$

where we have used $\|\vartheta - \theta^*_{\ell k(S_{\ell k})}\|_2 \leq \gamma$ and the groupwise irrepresentability condition in A1.

We also have the following two facts, which we will use momentarily:

$$\Phi_{\max}^3 n s \gamma^2 = o(\lambda_1) \tag{S.14}$$

$$\sqrt{s}\Phi_{\max}\gamma \to 0. \tag{S.15}$$

Note that (S.14) can be obtained as follows. Since $(1/2)(x+y)^2 \leq x^2 + y^2$ for $x, y \in \mathbf{R}$, we can plug in

$$\gamma = C\left(\frac{\lambda_1 s\sqrt{m}}{n} + \sqrt{\frac{s\log n}{n}}\right),$$

and check that both terms on the right-hand side of

$$\frac{\Phi_{\max}^3 n s}{\lambda_1}\left(\frac{\lambda_1^2 s^2 m}{n^2} + \frac{s\log n}{n}\right) = \frac{\Phi_{\max}^3 s^3 \lambda_1 m}{n} + \frac{\Phi_{\max}^3 s^2 \log n}{\lambda_1}$$

tend to zero. For the first term on the right-hand side, it is enough to show that

$$\Phi_{\max}^6 s^6 m^3 \log(d^2 mr)(\log^3 n)/n \to 0,$$

where we have plugged in $\lambda_1 = C'\sqrt{mn\log(d^2mr/\delta)\log^3 n}$. Using the assumptions in condition A4, we get that $\log(d^2mr) = O(\log d + \log m) = O(n^{2/21})$, and furthermore that

$$\Phi_{\max}^6 s^6 m^3 \log(d^2 mr)(\log^3 n)/n = o\left(\frac{n^{1/3} \cdot n^{2/21} \cdot n^{6/21} \cdot n^{6/21}}{\log^3 n}\right)\frac{\log^3 n}{n} \to 0,$$

as required. A similar calculation shows that the second term on the right-hand side also tends to zero, *i.e.*, $\Phi_{\max}^3 s^2 (\log n)/\lambda_1 \to 0$, which establishes (S.14). Lastly, (S.15) follows since its left-hand side is dominated by the left-hand side of (S.14).

So, we now compute

$$\|\Delta_4^j\|_2 \leq \sqrt{m}\max_t \sum_{i=1}^n \left|\Phi_{it}\left[F_{\epsilon_{\ell k}}\left(\Phi_{i(S_{\ell k})}\left(\vartheta - \theta^*_{\ell k(S_{\ell k})}\right)\right) - \right.\right.$$

$$\left.\left. F_{\epsilon_{\ell k}}(0) - f_{\epsilon_{\ell k}}(0)\Phi_{i(S_{\ell k})}\left(\vartheta - \theta^*_{\ell k(S_{\ell k})}\right)\right]\right|$$

$$\leq C_1 \Phi_{\max}\sqrt{m}\sum_{i=1}^n \left(\Phi_{i(S_{\ell k})}\left(\vartheta - \theta^*_{\ell k(S_{\ell k})}\right)\right)^2$$

$$\leq C_1 \Phi_{\max} \sqrt{m} \sum_{i=1}^{n} \|\Phi_{i(S_{\ell k})}\|_2^2 \|\vartheta - \theta^*_{\ell k(S_{\ell k})}\|_2^2$$
$$\leq C_1 \Phi_{\max}^3 \sqrt{m} n s \gamma^2$$
$$= o(\lambda_1).$$

Here the first inequality follows from the fact that $\|x\|_2 \leq \sqrt{m}\|x\|_\infty$ for $x \in \mathbf{R}^m$, and the triangle inequality; the second follows from the distributional smoothness condition in A2, which is applicable since (S.15) holds; the third uses Cauchy-Schwarz; the fourth uses our column norm assumption on $\Phi$, and $\|\vartheta - \theta^*_{\ell k(S_{\ell k})}\|_2 \leq \gamma$; the last uses (S.14). As $\|\Delta_4^j\|_2 = o(\lambda_1)$, it will certainly be strictly less than $\lambda_1/6$ for $n$ large enough. We have hence shown, noting that none of our above arguments have depended on the particular choice of $j = 1, \dots, d$ or $\vartheta \in B^*$,

$$\max_{j \in S_{\ell k}^c} \sup_{\vartheta \in B^*} \|\Delta_2^j\|_2 < \frac{\lambda_1}{3}.$$

*Bounding* $\|\Delta_3^j\|_2$. For this part, we can use the end of the proof of Lemma 2 in [3], which uses classic entropy-based techniques to establish a bound very similar to that which we are seeking. By carefully looking at the conditions required for this lemma, we see that under the distributional smoothness condition in A2, condition A3, and also

$$\sqrt{n \log(dm)} = o(\lambda_1)$$
$$n \Phi_{\max} \gamma^2 = o(\lambda_1)$$
$$(1 + \gamma \Phi_{\max}^2 s^{3/2}) \log^2 n = o(\lambda_1^2/n),$$

all following directly from condition A4 by calculations similar to the ones we used when bounding $\|\Delta_2^j\|$, we have

$$\mathbf{Pr}\left( \max_{j \in S_{\ell k}^c} \sup_{\vartheta \in B^*} \|\Delta_3^j\|_2 \geq \frac{\lambda_1}{3} \right) \leq \mathbf{Pr}\left( \max_{j \in S_{\ell k}^c} \sup_{\vartheta \in B^*} \|\Delta_3^j\|_\infty \geq \frac{\lambda_1}{3\sqrt{m}} \right);$$

the probability on the right-hand side can be made arbitrarily small for large $n$, by the arguments at the end of Lemma 2 in [3], and hence clearly smaller than the desired $\delta/(4dr)$ level.

*Putting it together.* Returning to the logic in (S.11), (S.12), (S.13), we have shown that the subgradient condition in (S.11) holds with probability at least $1 - (\delta/(2dr) + \delta/(4dr) + \delta/(4dr)) = 1 - \delta/(dr)$. Taking a union bound over $k = 1, \dots, d$ and $\ell = 1, \dots, r$, which were considered fixed at the start of our analysis, gives the result stated in the theorem. $\qquad\square$

## S.5  Statement and proof of Lemma S.5.1

We show that with probability at least $1 - \delta/(2dr)$, it holds that $\tilde{\theta}_{\ell k(S_{\ell k})} \in B^*$, where $\tilde{\theta}_{\ell k(S_{\ell k})}$ is the solution to the restricted problem (S.8), for some fixed $k = 1, \dots, d$ and $\ell = 1, \dots, r$, and $B^*$ is a ball defined in the proof of Theorem 4.3 in Section S.4. This fact is used a few times in the proof of Theorem 4.3.

**Lemma S.5.1.** *Fix some* $k = 1, \dots, d$ *and* $\ell = 1, \dots, r$. *Define the ball*

$$B^* = \{\vartheta \in \mathbf{R}^{sm} : \|\vartheta - \theta^*_{\ell k(S_{\ell k})}\|_2 \leq \gamma\}$$

*centered at the underlying coefficients* $\theta^*_{\ell k(S_{\ell k})}$ *with radius*

$$\gamma = C\left( \frac{\lambda_1 s \sqrt{m}}{n} + \sqrt{\frac{s \log n}{n}} \right),$$

*for some constant* $C > 0$. *Then, with probability at least* $1 - \delta/(2dr)$, *it holds that* $\tilde{\theta}_{\ell k(S_{\ell k})} \in B^*$, *where* $\tilde{\theta}_{\ell k(S_{\ell k})}$ *is the solution to the restricted problem* (S.8).

*Proof.* We will follow the strategy for the proof of Theorem 1 in [3] closely. We begin by considering the ball

$$B = \{\vartheta \in \mathbf{R}^{sm} : \|\vartheta - \theta^*_{\ell k(S_{\ell k})}\|_2 \leq R\}$$

with center $\theta^*_{\ell k(S_{\ell k})}$ and radius $R$. We also introduce some useful notational shorthand, and write the quantile loss term in the restricted problem (S.8) as

$$L_{\ell k}(\vartheta) = \psi_{\alpha_\ell} \left( Y_k - \Phi_{S_{\ell k}} \vartheta \right).$$

Below, we show that a particular function of $R$ serves as an upper bound for the quantity $\mathbf{E}[L_{\ell k}(\tilde{\vartheta}_{\ell k(S_{\ell k})}) - L_{\ell k}(\theta^*_{\ell k(S_{\ell k})})]$, where the expectation here is taken over draws of the data, and $\tilde{\vartheta}_{\ell k(S_{\ell k})}$ is a particular point in $B$ that we define in a moment. This in turn implies, with probability at least $1 - \delta/(2dr)$, that $\tilde{\theta}_{\ell k(S_{\ell k})} \in B^*$, as claimed.

First, we define $\tilde{\vartheta}_{\ell k(S_{\ell k})}$ more precisely: it is a point on the line segment between the solution to the restricted problem $\tilde{\theta}_{\ell k(S_{\ell k})}$ and the underlying coefficients $\theta^*_{\ell k(S_{\ell k})}$, *i.e.*,

$$\tilde{\vartheta}_{\ell k(S_{\ell k})} = \beta \tilde{\theta}_{\ell k(S_{\ell k})} + (1 - \beta) \theta^*_{\ell k(S_{\ell k})},$$

for a particular choice

$$\beta = \frac{R}{R + \|\tilde{\theta}_{\ell k(S_{\ell k})} - \theta^*_{\ell k(S_{\ell k})}\|_2},$$

which guarantees that $\tilde{\vartheta}_{\ell k(S_{\ell k})} \in B$ even if $\tilde{\theta}_{\ell k(S_{\ell k})} \notin B$ , as we establish next. Observe that we always have

$$\|\tilde{\theta}_{\ell k(S_{\ell k})} - \theta^*_{\ell k(S_{\ell k})}\|_2 \leq R + \|\tilde{\theta}_{\ell k(S_{\ell k})} - \theta^*_{\ell k(S_{\ell k})}\|_2$$

$$\iff R \frac{\|\tilde{\theta}_{\ell k(S_{\ell k})} - \theta^*_{\ell k(S_{\ell k})}\|_2}{R + \|\tilde{\theta}_{\ell k(S_{\ell k})} - \theta^*_{\ell k(S_{\ell k})}\|_2} \leq R$$

$$\iff \beta \|\tilde{\theta}_{\ell k(S_{\ell k})} - \theta^*_{\ell k(S_{\ell k})}\|_2 \leq R$$

$$\iff \|\beta \tilde{\theta}_{\ell k(S_{\ell k})} - \beta \theta^*_{\ell k(S_{\ell k})} + \theta^*_{\ell k(S_{\ell k})} - \theta^*_{\ell k(S_{\ell k})}\|_2 \leq R$$

$$\iff \|\tilde{\vartheta}_{\ell k(S_{\ell k})} - \theta^*_{\ell k(S_{\ell k})}\|_2 \leq R,$$

as claimed. The second line here follows by rearranging and multiplying through by $R$; the third by using the definition of $\beta$ above; the fourth by adding and subtracting the underlying coefficients; and the fifth by using the definition of $\tilde{\vartheta}_{\ell k(S_{\ell k})}$.

Now, the beginning of the proof of Theorem 1 in [3] establishes, for any $\tilde{\vartheta}_{\ell k(S_{\ell k})} \in B$, for some constant $C_5 > 0$, and using condition A3, that

$$\mathbf{E}\left[ L_{\ell k}(\tilde{\vartheta}_{\ell k(S_{\ell k})}) - L_{\ell k}(\theta^*_{\ell k(S_{\ell k})}) \right] \geq C_5 n \|\tilde{\vartheta}_{\ell k(S_{\ell k})} - \theta^*_{\ell k(S_{\ell k})}\|_2^2, \tag{S.16}$$

and so, by direct calculation, since

$$\|\tilde{\vartheta}_{\ell k(S_{\ell k})} - \theta^*_{\ell k(S_{\ell k})}\|_2 \leq R \iff \beta \|\tilde{\theta}_{\ell k(S_{\ell k})} - \theta^*_{\ell k(S_{\ell k})}\|_2 \leq R \iff \|\tilde{\theta}_{\ell k(S_{\ell k})} - \theta^*_{\ell k(S_{\ell k})}\|_2 \leq R/2, \tag{S.17}$$

it suffices to obtain a suitable upper bound for $\mathbf{E}[L_{\ell k}(\tilde{\vartheta}_{\ell k(S_{\ell k})}) - L_{\ell k}(\theta^*_{\ell k(S_{\ell k})})]$, in order to get the result in the statement of the lemma. To this end, we introduce one more piece of shorthand, and denote the objective for the restricted problem (S.8) as $J_{\ell k}(\vartheta)$.

We proceed with the following chain of (in)equalities:

$$\mathbf{E}\left[ L_{\ell k}(\tilde{\vartheta}_{\ell k(S_{\ell k})}) - L_{\ell k}(\theta^*_{\ell k(S_{\ell k})}) \right]$$

$$= \mathbf{E}\left[ L_{\ell k}(\tilde{\vartheta}_{\ell k(S_{\ell k})}) - L_{\ell k}(\theta^*_{\ell k(S_{\ell k})}) \right] + J_{\ell k}(\tilde{\vartheta}_{\ell k(S_{\ell k})}) - J_{\ell k}(\tilde{\vartheta}_{\ell k(S_{\ell k})}) +$$

$$J_{\ell k}(\theta^*_{\ell k(S_{\ell k})}) - J_{\ell k}(\theta^*_{\ell k(S_{\ell k})}) \tag{S.18}$$

$$= \underbrace{L_{\ell k}(\theta^*_{\ell k(S_{\ell k})}) - \mathbf{E}\, L_{\ell k}(\theta^*_{\ell k(S_{\ell k})}) - L_{\ell k}(\tilde{\vartheta}_{\ell k(S_{\ell k})}) + \mathbf{E}\, L_{\ell k}(\tilde{\vartheta}_{\ell k(S_{\ell k})})}_{\Delta(\theta^*_{\ell k(S_{\ell k})}, \tilde{\vartheta}_{\ell k(S_{\ell k})})} +$$

$$J_{\ell k}(\tilde{\vartheta}_{\ell k(S_{\ell k})}) - J_{\ell k}(\theta^*_{\ell k(S_{\ell k})}) + \lambda_1 \sum_{j \in S_{\ell k}} \|\theta^*_{\ell k j}\|_2 - \lambda_1 \sum_{j \in S_{\ell k}} \|\tilde{\vartheta}_{\ell k j}\|_2$$
$$- (\lambda_2/2)\|\tilde{\vartheta}_{\ell k(S_{\ell k})}\|_2^2 + (\lambda_2/2)\|\theta^*_{\ell k(S_{\ell k})}\|_2^2 \tag{S.19}$$

$$\leq \Delta(\theta^*_{\ell k(S_{\ell k})}, \tilde{\vartheta}_{\ell k(S_{\ell k})}) + J_{\ell k}(\tilde{\vartheta}_{\ell k(S_{\ell k})}) - J_{\ell k}(\theta^*_{\ell k(S_{\ell k})}) + \lambda_1 \sum_{j \in S_{\ell k}} \|\theta^*_{\ell k j} - \tilde{\vartheta}_{\ell k j}\|_2 + o(1) \tag{S.20}$$

$$\leq \Delta(\theta^*_{\ell k(S_{\ell k})}, \tilde{\vartheta}_{\ell k(S_{\ell k})}) + J_{\ell k}(\tilde{\vartheta}_{\ell k(S_{\ell k})}) - J_{\ell k}(\theta^*_{\ell k(S_{\ell k})}) + \lambda_1 s R\sqrt{m} + o(1) \tag{S.21}$$

$$\leq \Delta(\theta^*_{\ell k(S_{\ell k})}, \tilde{\vartheta}_{\ell k(S_{\ell k})}) + 2\lambda_1 s R\sqrt{m} \tag{S.22}$$

$$\leq \sup_{\tilde{\vartheta}_{\ell k(S_{\ell k})} \in B} |\Delta(\theta^*_{\ell k(S_{\ell k})}, \tilde{\vartheta}_{\ell k(S_{\ell k})})| + 2\lambda_1 s R\sqrt{m}. \tag{S.23}$$

Here, (S.18) follows by adding and subtracting like terms, and (S.19) by rearranging (S.18). In (S.20) we use the triangle inequality and the following argument to show that the terms involving $\lambda_2$ are $o(1)$. Under the assumption that $\lambda_2 = o(n^{41/42}/\theta^*_{\max})$, combined with the restriction that $s = o(n^{1/21})$, we have $\lambda_2 = o(n/(\sqrt{s}\theta^*_{\max}))$. Therefore, under our choice of $R = 1/n$ (as specified below), we have

$$\lambda_2 \sqrt{s}\theta^*_{\max} R \to 0.$$

This in turn is used to argue that

$$-(\lambda_2/2)\|\tilde{\vartheta}_{\ell k(S_{\ell k})}\|_2^2 + (\lambda_2/2)\|\theta^*_{\ell k(S_{\ell k})}\|_2^2 = (\lambda_2/2)\|\tilde{\vartheta}_{\ell k(S_{\ell k})} - \theta^*_{\ell k(S_{\ell k})}\|_2^2$$
$$- \lambda_2\|\tilde{\vartheta}_{\ell k(S_{\ell k})}\|_2^2 + \lambda_2 \tilde{\vartheta}^T_{\ell k(S_{\ell k})}\theta^*_{\ell k(S_{\ell k})}$$
$$\leq (\lambda_2/2)R^2 - \lambda_2\|\tilde{\vartheta}_{\ell k(S_{\ell k})}\|_2(\|\tilde{\vartheta}_{\ell k(S_{\ell k})}\|_2 - \|\theta^*_{\ell k(S_{\ell k})}\|_2)$$
$$\leq (\lambda_2/2)R^2 + \lambda_2\|\tilde{\vartheta}_{\ell k(S_{\ell k})}\|_2 R$$
$$\leq (\lambda_2/2)R^2 + \lambda_2\|\theta^*_{\ell k(S_{\ell k})}\|_2 R$$
$$\leq (\lambda_2/2)R^2 + \lambda_2 \sqrt{s}\theta^*_{\max} R \to 0.$$

In the second to last line, we have applied $\|\tilde{\vartheta}_{\ell k(S_{\ell k})}\|_2 \leq \|\theta^*_{\ell k(S_{\ell k})}\|_2$, as, outside of this case, the term in question $-(\lambda_2/2)\|\tilde{\vartheta}_{\ell k(S_{\ell k})}\|_2^2 + (\lambda_2/2)\|\theta^*_{\ell k(S_{\ell k})}\|_2^2$ would be negative, anyway.

Continuing on, (S.21) holds because $\|\theta^*_{\ell k(S_{\ell k})} - \tilde{\vartheta}_{\ell k(S_{\ell k})}\|_2 \leq R$ implies $\|\theta^*_{\ell k j} - \tilde{\vartheta}_{\ell k j}\|_2 \leq R$. Finally, (S.22) follows because of the following argument. Since $J_{\ell k}$ is convex, we can use the definition of $\tilde{\vartheta}_{\ell k(S_{\ell k})}$ and get

$$J_{\ell k}(\tilde{\vartheta}_{\ell k(S_{\ell k})}) \leq \beta J_{\ell k}(\tilde{\theta}_{\ell k(S_{\ell k})}) + (1-\beta)J_{\ell k}(\theta^*_{\ell k(S_{\ell k})}) = J_{\ell k}(\theta^*_{\ell k(S_{\ell k})}) + \beta(J_{\ell k}(\tilde{\theta}^*_{\ell k(S_{\ell k})}) - J_{\ell k}(\theta^*_{\ell k(S_{\ell k})}));$$

notice that the last term here is nonpositive, since $\tilde{\theta}_{\ell k(S_{\ell k})}$ is the solution to the restricted problem (S.8), and thus we have that

$$J_{\ell k}(\tilde{\vartheta}_{\ell k(S_{\ell k})}) \leq J_{\ell k}(\theta^*_{\ell k(S_{\ell k})}),$$

which lets us move from (S.21) to (S.22).

Lemma 1 in [3] states, with probability at least $1 - \delta$, where $\delta = \exp(-C_6 s \log n)$ and $C_6 > 0$ is some constant, that

$$\sup_{\tilde{\vartheta}_{\ell k(S_{\ell k})} \in B} |\Delta(\theta^*_{\ell k(S_{\ell k})}, \tilde{\vartheta}_{\ell k(S_{\ell k})})| \leq 6R\sqrt{sn \log n},$$

so from (S.23), with probability at least $1 - \delta$, we see that

$$\mathbf{E}\left[L_{\ell k}(\tilde{\vartheta}_{\ell k(S_{\ell k})}) - L_{\ell k}(\theta^*_{\ell k(S_{\ell k})})\right] \leq 6R\sqrt{sn \log n} + 2\lambda_1 s R\sqrt{m}$$

and, using (S.16), that

$$n\|\tilde{\vartheta}_{\ell k(S_{\ell k})} - \theta^*_{\ell k(S_{\ell k})}\|_2^2 \leq C' \left( R\sqrt{sn\log n} + \lambda_1 sR\sqrt{m} \right),$$

for some constant $C' > 0$.

Plugging in $R = 1/n$, dividing through by $n$, and using the fact that the square root function is subadditive, we get, with probability at least $1 - \delta$, that

$$\|\tilde{\vartheta}_{\ell k(S_{\ell k})} - \theta^*_{\ell k(S_{\ell k})}\|_2 \leq C' \left( \frac{(s\log n)^{1/4}}{n^{3/4}} + \frac{(\lambda_1 s)^{1/2} m^{1/4}}{n} \right)$$
$$\leq C' \left( \sqrt{\frac{s\log n}{n}} + \frac{\lambda_1 s\sqrt{m}}{n} \right).$$

Finally, we complete the proof by applying (S.17), in order to get that

$$\|\tilde{\theta}_{\ell k(S_{\ell k})} - \theta^*_{\ell k(S_{\ell k})}\|_2 \leq \gamma,$$

where we have defined

$$\gamma = C \left( \frac{\lambda_1 s\sqrt{m}}{n} + \sqrt{\frac{s\log n}{n}} \right),$$

and $C > 0$ is some constant, with probability at least $1 - \delta/(2dr)$, for large enough $n$. $\qquad\square$

## S.6 Proof of Lemma 5.1

The prox operator $\mathbf{prox}_{\lambda\psi_{\mathcal{A}}}(A)$ is separable in the entries of its minimizer $X$, so we can focus on minimizing over $X_{ij}$ the expression

$$\max\{\alpha_j X_{ij}, (\alpha_j - 1)X_{ij}\} + (1/(2\lambda))\left(X_{ij} - A_{ij}\right)^2$$
$$= \alpha_j \max\{0, X_{ij}\} + (1 - \alpha_j)\max\{0, -X_{ij}\} + (1/(2\lambda))\left(X_{ij} - A_{ij}\right)^2. \qquad \text{(S.24)}$$

Suppose $X_{ij} > 0$. Then differentiating (S.24) gives $X_{ij} = A_{ij} - \lambda\alpha_j$ and the sufficient condition $A_{ij} > \lambda\alpha_j$. Similarly, assuming $X_{ij} < 0$ gives $X_{ij} = A_{ij} + \lambda(1 - \alpha_j)$ when $A_{ij} < \lambda(\alpha_j - 1)$. Otherwise, we can take $X_{ij} = 0$. Putting these cases together gives the result. $\qquad\square$

## S.7 ADMM for the MQGM

A complete description of our ADMM-based algorithm for fitting the MQGM to data is given in Algorithm 1.

## S.8 Additional details on Gibbs sampling

In the MQGM, there is no analytic solution for parameters like the mean, median, or quantiles of these marginal and conditional distributions, but the pseudolikelihood approximation makes for a very efficient Gibbs sampling procedure, which we highlight in this section. As it is relevant to the computational aspects of the approach, in this subsection we will make explicit the conditional random field, where $y_k$ depends on both $y_{\neg k}$ and fixed input features $x$.

First, note that since we are representing the distribution of $y_k|y_{\neg k}, x$ via its inverse CDF, to sample from from this conditional distribution we can simply generate a random $\alpha \sim \text{Uniform}(0, 1)$. We then compute

$$\hat{Q}_{y_k|y_{\neg k}}(\alpha_\ell) = \phi(y)^T \theta_{\ell k} + x^T \theta^x_{\ell k}$$
$$\hat{Q}_{y_k|y_{\neg k}}(\alpha_{\ell+1}) = \phi(y)^T \theta_{(\ell+1)k} + x^T \theta^x_{(\ell+1)k}$$

for some pair $\alpha_\ell \leq \alpha \leq \alpha_{\ell+1}$ and set $y_k$ to be a linear interpolation of the two values,

---

**Algorithm 1** ADMM for the MQGM

---

**Input:** observations $y^{(1)}, \ldots, y^{(n)} \in \mathbf{R}^d$, feature matrix $\Phi \in \mathbf{R}^{n \times dm}$, quantile levels $\mathcal{A}$, constants $\lambda_1, \lambda_2 > 0$
**Output:** fitted coefficients $\hat{\Theta} = (\hat{\theta}_{\ell k j}, \hat{b}_{\ell k})$
**for** $k = 1, \ldots, d$ (in parallel, if possible) **do**
    **initialize** $\Theta_k, B_k, V, W, Z, U_V, U_W, U_Z$
    **repeat**
        update $\Theta_k$ using (13)
        update $B_k$ using (13)
        update $V$ using (10)
        update $W$ using (11)
        update $Z$ using (12) and Lemma 5.1
        update $U_V, U_W, U_Z$:

$$U_V \leftarrow U_V + (\mathbf{1}B_k^T + \Phi_k\Theta - V)$$
$$U_W \leftarrow U_W + (\Theta_k - W)$$
$$U_Z \leftarrow U_Z + (Y_k\mathbf{1}^T - \mathbf{1}B_k^T - \Phi_k\Theta - Z)$$

    **until** converged
  **end for**

---

$$y_k \leftarrow \hat{Q}_{y_k|y_{\neg k}}(\alpha_\ell) + \frac{\left(\hat{Q}_{y_k|y_{\neg k}}(\alpha_{\ell+1}) - \hat{Q}_{y_k|y_{\neg k}}(\alpha_\ell)\right)(\alpha - \alpha_\ell)}{\alpha_{\ell+1} - \alpha_\ell}.$$

This highlights the desirability of having a range of nonuniformly spaced $\alpha$ terms that reach values close to zero and one as otherwise we may not be able to find a pair of $\alpha$'s that lower and upper bound our random sample $\alpha$. However, in the case that we model a sufficient quantity of $\alpha$, a reasonable approximation (albeit one that will not sample from the extreme tails) is also simply to pick a random $\alpha_\ell \in \mathcal{A}$ and use just the corresponding column $\theta_{\ell k}$ to generate the random sample.

Computationally, there are a few simple but key points involved in making the sampling efficient. First, when sampling from a conditional distribution, we can precompute $x^T\Theta_k^x$ for each $k$, and use these terms as a constant offset. Second, we maintain a "running" feature vector $\phi(y) \in \mathbf{R}^{dm}$, *i.e.*, the concatenation of features corresponding to each coordinate $\phi(y_k)$. Each time we sample a new coordinate $y_k$, we generate just the new features in the $\phi(y_k)$ block, leaving the remaining features untouched. Finally, since the $\Theta_k$ terms are sparse, the inner product $\phi(y)^T\theta_{\ell k}$ will only contain a few nonzeros terms in the sum, and will be computed more efficiently if the $\Theta_k$ are stored as a sparse matrices.

## S.9    Additional numerical results for the ring data

### S.9.1    Conditional independencies recovered by the nonparanormal skeptic, TIGER, and Laplace

We present the conditional independencies recovered by the nonparanormal skeptic, TIGER, and Laplace on the ring data in Figure S.1; results for the remaining methods are presented in Section 6.1.

### S.9.2    Evaluation of fitted conditional CDFs

Here, we elaborate on the evaluation of the conditional CDFs given by the MQGM, MB, GLasso, SpaceJam, TIGER, and Laplace, when run on the ring data. (We omit the nonparanormal skeptic from our evaluation as it is not clear how to sample from its conditionals, due to the nature of a particular transformation that it uses.)

For each of these methods, we essentially averaged the total variation distances and Kolmogorov-Smirnoff statistics between the fitted and true conditional CDFs across all variables, and then reported

**Figure S.1:** Conditional independencies recovered by the nonparanormal skeptic, TIGER, and Laplace on the ring data; black means conditional dependence.

**Table S.1:** Total variation (TV) distance and Kolmogorov-Smirnoff (KS) statistic values for the MQGM, MB, GLasso, SpaceJam, TIGER, and Laplace on the ring data; lower is better, best in **bold**.

|          | TV        | KS        |
|----------|-----------|-----------|
| MQGM     | **20.873**| **0.760** |
| MB       | 92.298    | 1.856     |
| GLasso   | 92.479    | 1.768     |
| SpaceJam | 91.568    | 1.697     |
| TIGER    | 88.284    | 1.450     |
| Laplace  | 127.406   | 1.768     |

the best values obtained across a range of tuning parameters (more details below). We present the results in Table S.1; we see that the MQGM outperforms all its competitors, in both metrics.

We now describe the evaluation in more detail; for simplicity, we describe everything that follows in terms of the conditional CDF $y_1|y_2$ only, with everything being extended in the obvious way to other conditionals.

First, we carried out the following steps in order to compute the true (empirical) conditional CDF.

1. We drew $n = 400$ samples from the ring distribution, by following the procedure described in Section 6.1; these observations are plotted across the top row of Figure S.2.

2. We then partitioned the $y_2$ samples into five equally-sized bins, and computed the true empirical conditional CDF of $y_1$ given each bin of $y_2$ values.

Next, we carried out the following steps in order to compute the estimated (empirical) conditional CDFs, for each method.

3. We fitted each method to the samples obtained in step (1) above.

4. Then, for each method, we drew a sample of $y_1$ given *each* $y_2$ sample, using the method's conditional distribution; these conditionals are plotted across the second through fifth rows of Figure S.2 (for representative values of $\lambda_1$).

   Operationally, we drew samples from each method's conditionals in the following ways.

   • MQGM: we used the Gibbs sampler described in Section S.8.

   • MB: we drew $y_1 \sim \mathcal{N}(\hat{\theta}_1^T y_2^{(i)}, \hat{\sigma}_{1|2}^2)$, where $\hat{\theta}_1$ is the fitted lasso regression coefficient of $y_1$ on $y_2$; $y_2^{(i)}$ for $i = 1, \ldots, n$ is the $i$th observation of $y_2$ obtained in step (1) above; and $\hat{\sigma}_{1|2}^2 = \mathbf{var}(Y_1 - Y_2\hat{\theta}_1)$ denotes the sample variance of the underlying error term $Y_1 - Y_2\hat{\theta}_1$ with $Y_i = (y_i^{(1)}, \ldots, y_i^{(n)}) \in \mathbf{R}^n$ collecting all observations along variable $i$.

- SpaceJam: we drew $y_1 \sim \mathcal{N}(\hat{\theta}_1^T \phi(y_2^{(i)}), \hat{\sigma}_{1|2}^2)$, where $\phi$ is a suitable basis function, and $\hat{\theta}_1$ as well as $\hat{\sigma}_{1|2}^2$ are defined in ways analogous to the neighborhood selection setup.
- GLasso: we drew $y_1 \sim \mathcal{N}(\hat{\mu}_{1|2}, \hat{\sigma}_{1|2}^2)$, where

$$\hat{\mu}_{1|2} = \hat{\mu}_1 + \hat{\Sigma}_{12}\hat{\Sigma}_{22}^{-1}(y_2^{(i)} - \hat{\mu}_2)$$
$$\hat{\sigma}_{1|2}^2 = \hat{\Sigma}_{11} - \hat{\Sigma}_{12}\hat{\Sigma}_{22}^{-1}\hat{\Sigma}_{21}$$

  with $\hat{\mu}_i$ denoting the sample mean of $Y_i$, and $\hat{\Sigma}$ denoting the estimate of the covariance matrix given by GLasso (subscripts select blocks of this matrix).

5. Finally, we partitioned the $y_2$ samples into five equally-sized bins (just as when computing the true conditional CDF), and computed the estimated empirical conditional CDF of $y_1$ given each bin of $y_2$ values.

Having computed the true as well as estimated conditional CDFs, we measured the goodness of fit of each method's conditional CDFs to the true conditional CDFs, by computing the total variation (TV) distance, *i.e.*,

$$(1/2)\sum_{i=1}^{q} \left| \hat{F}_{y_1|y_2}^{\mathrm{method}_j}(z^{(i)}|\zeta) - \hat{F}_{y_1|y_2}^{\mathrm{true}}(z^{(i)}|\zeta) \right|,$$

as well as the (scaled) Kolmogorov-Smirnoff (KS) statistic, *i.e.*,

$$\max_{i=1,\dots,q} \left| \hat{F}_{y_1|y_2}^{\mathrm{method}_j}(z^{(i)}|\zeta) - \hat{F}_{y_1|y_2}^{\mathrm{true}}(z^{(i)}|\zeta) \right|.$$

Here, $\hat{F}_{y_1|y_2}^{\mathrm{true}}(z^{(i)}|\zeta)$ is the true empirical conditional CDF of $y_1|y_2$, evaluated at $y_1 = z^{(i)}$ and given $y_2 = \zeta$, and $\hat{F}_{y_1|y_2}^{\mathrm{method}_j}(z^{(i)}|\zeta)$ is a particular method's ("method$_j$" above) estimated empirical conditional CDF, evaluated at $y_1 = z^{(i)}$ and given $y_2 = \zeta$. For each method, we averaged these TV and KS values across the method's conditional CDFs. Table S.1 reports the best (across a range of tuning parameters) of these averaged TV and KS values.

## S.10 Additional numerical results for modeling flu epidemics

Here, we plot samples from the marginal distributions of the percentages of flu reports at regions one, five, and ten throughout the year, which reveals the heteroskedastic nature of the data (just as in Section 6.2, for region six).

## S.11 Sustainable energy application

We evaluate the ability of MQGM to recover the conditional independencies between several wind farms on the basis of large-scale, hourly wind power measurements; wind power is intermittent, and thus understanding the relationships between wind farms can help farm operators plan. We obtained hourly wind power measurements from July 1, 2009 through September 14, 2010 at seven wind farms ($n = 877$, see [6, 8, 1] for details). The primary variables here encode the hourly wind power at a farm over two days (*i.e.*, 48 hours), thus $d = 7 \times 48 = 336$. Exogenous variables were used to encode forecasted wind power and direction as well as other historical measurements, for a total of $q = 3417$. We set $m = 5$ and $r = 20$. Fitting the MQGM here hence requires solving $48 \times 7 = 336$ multiple quantile regression subproblems each of dimension $((336 - 1) \times 5 + 3417) \times 20 = 101,840$. Each subproblem took roughly 87 minutes, comparable to the algorithm of [8].

Figure S.4 presents the recovered conditional independencies; the nonzero super- and sub-diagonal entries suggest that at any wind farm, the previous hour's wind power (naturally) influences the next hour's, while the nonzero off-diagonal entries, *e.g.*, in the (4,6) block, uncover farms that may influence one another. [8], whose method placed fifth in a Kaggle competition, as well as [1] report similar findings (see the left panels of Figures 7 and 3 in these papers, respectively).

**Figure S.2:** Conditional distributions for MQGM, MB, GLasso, and SpaceJam, fitted to samples from the ring distribution (TIGER and Laplace's conditionals both look similar to MB's). First row: samples from the ring distribution, where each plot highlights the samples falling into a particular shaded bin on the $y_2$ axis. Second through fifth rows: conditional distributions of $y_1$ given $y_2$ for each method, where each plot conditions on the appropriate $y_2$ bin as highlighted in the first row. The MQGM's conditional distributions are intuitive, appearing bimodal for bin 3, and more peaked for bins 1 and 5. MB, GLasso, and SpaceJam's densities appear (roughly) Gaussian, as expected.

**Figure S.3:** Samples from the fitted marginal distributions of the weekly flu incidence rates at several regions of the U.S.; samples at larger quantile levels shaded lighter, median in darker blue.

**Figure S.4:** Conditional independencies recovered by the MQGM on the wind farms data; each block corresponds to a wind farm, and black indicates dependence.