[Reviews · NeurIPS 2016]

Reviewer 1

Summary

This paper studied the estimation of sparse graphical models in the high-dimensional setting. Instead of modeling the conditional dependencies among the random variables under some parametric assumptions, the authors proposed to models the ‚Äúconditional dependencies‚ÄĚ using existing techniques in the quantile regression literature. I find the paper clearly written and easy to follow. Simulation studies are compared to several existing graphical modeling approaches and the numerical results seem to be promising.

Qualitative Assessment

(1) It seems to me that this is a direct application of existing quantile regression techniques to graphical modeling. In terms of methodology, is there anything novel here compared to quantile regression? In terms of theory, the authors mentioned that it is a adaptation of Theorem 2 in Fan et al (AOS 2014). To justify a publication in NIPS, can the authors clarify the main contribution of this paper? (2) As the authors mentioned in Section 4.2, there might not exists a joint distribution to characterize the usual notion of conditional dependencies in pairwise graphical models. I find this fact very concerning since a graph cannot be defined without a joint density. (3) What is the computational complexity of the ADMM algorithm? Can coordinate descent be used instead? (4) There are a lot more flexible graphical modeling methodologies that are proposed recently. For instance, Yang et al (arXiv 2014) on semi parametric exponential family graphical models. The authors should compare these methods in the numerical studies section.

Confidence in this Review

2-Confident (read it all; understood it all reasonably well)


Reviewer 2

Summary

This paper details a general neighborhood selection approach to estimating Graphical models. This approach models a set of conditional quantiles for each node conditioned on the rest. This enables estimation of the graph even in cases of non-Gaussian/ heavy tailed data. The authors propose an ADMM algorithm for estimation of the MQG models. They also provide statistical guarantees for structure recovery by MQGM in graphical models.

Qualitative Assessment

The paper considers a very interesting approach to estimating Graphical models. I certainly believe that this approach will lead to a promising direction of research. I have the following comments: 1. In subsection 4.3, for the graph structure recovery, the conditional dependence/ edge set E star is defined only through the "r" quantiles. I wonder how realistic this assumption is; does it require that r must be large in order for it to be a good approximation to true dependence. Why not define the absence of edges in terms of true conditional independence for all alpha in zero and one. 2. Could the authors provide some discussions in Section 4.2 as to why the estimation of joint distribution is important in terms of estimation of graphical models? 3. I believe the equation on the first line after equation (5) (Line 116) should be psi instead of phi.

Confidence in this Review

2-Confident (read it all; understood it all reasonably well)


Reviewer 3

Summary

The paper presents the multiple quantile graphical models, which can be regarded as extension of the dependency networks to the continuous case. The proposed model can be used to model the joint relations amongst heteroskedastic non-Gaussian variables. The paper presents algorithms for such a model and the empirical results demonstrate the effectiveness of the model.

Qualitative Assessment

I think the presentation of this paper is very clear. The discussions are at the appropriate technical level. For someone who does not work on Gaussian graphical models, the paper is quite self-contained for me to understand the relevant background and capture the key contributions.

Confidence in this Review

1-Less confident (might not have understood significant parts)


Reviewer 4

Summary

This paper studies the Multiple Quantile Graphical Model (MQGM), which is a multiple quantile regression for high-dimensional, non-parametric distributions. It offers a much richer conditional distribution estimators, based on multiple quantile levels. The authors established theoretical and empirical supports that the MQGM is capable of representing the exact conditional independencies, even for heteroskedastic, non-Gaussian variables.

Qualitative Assessment

In this paper titled "The Multiple Quantile Graphical Model", the authors introduced the MQGM, which offers a much broader class of conditional distribution estimates by introducing a set of quantile levels . One key contribution of this paper is that the proposed MQGM asymptotically identifies the exact conditional independencies under some conditions, as the size of the graph grows. There is some empirical results of the proposed algorithm and other alternatives. The strength of the paper: + an interesting and important problem + the idea is simple and sound + theoretical proof My concern is mainly for significance of the key idea. The proposed method is very natural, and at a high level, introducing a set of quantile levels and providing more expressive class of estimates , which is not new. The main novelty here might be using multiple quantiles for non-parametric, continuous data. However, it is hard to catch what are the main challenges (compared to related methods) and how to handle it. One suggestion is that intuition about the above point could be made far detailed. The overall presentation of the paper is good, but the presentation of the reference part is quite hard to follow. The review of relevant work is elegant, but that would be better if less relevant citations can be deleted. Also, it would even be better to follow if the brief description of the algorithm, called ADMM is given in the main paper. [Typos] line 190: correpsonding -> corresponding

Confidence in this Review

2-Confident (read it all; understood it all reasonably well)


Reviewer 5

Summary

The paper introduced the Multiple Quantile Graphical Model (MQGM) which extends the neighborhood selection method for learning sparse graphical models. Graph structure recovery result was established under regularity conditions. The paper also developed an efficient algorithm for fitting the MQGM using ADMM.

Qualitative Assessment

Overall a nice paper proposing a new method for estimating sparse graphical models, even for non-Gaussian data. My main comments are (1) Could you give some discussion on the performance of MQGM in the presence of outlying/corrupted data? (2) For Gaussian/non-Gaussian data, it would be interesting to compare the experimental results of MQGM with TIGER (Liu and Wang, 2012)/Nonparanormal Skeptic (Liu et al., 2012).

Confidence in this Review

2-Confident (read it all; understood it all reasonably well)